# Coupling fission and exit of RAB6 vesicles at Golgi hotspots through kinesin-myosin interactions

Stéphanie Miserey-Lenkei[1], Hugo Bousquet[1], Olena Pylypenko[2], Sabine Bardin[1], Ariane Dimitrov[1], Gaëlle Bressanelli[2], Raja Bonifay[2], Vincent Fraisier[3], Catherine Guillou[4], Cécile Bougeret[5], Anne Houdusse [2], Arnaud Echard [6] & Bruno Goud[1]

The actin and microtubule cytoskeletons play important roles in Golgi structure and function, but how they are connected remain poorly known. In this study, we investigated whether RAB6 GTPase, a Golgi-associated RAB involved in the regulation of several transport steps at the Golgi level, and two of its effectors, Myosin IIA and KIF20A participate in the coupling between actin and microtubule cytoskeleton. We have previously shown that RAB6–Myosin IIA interaction is critical for the fission of RAB6-positive transport carriers from Golgi/TGN membranes. Here we show that KIF20A is also involved in the fission process and serves to anchor RAB6 on Golgi/TGN membranes near microtubule nucleating sites. We provide evidence that the fission events occur at a limited number of hotspots sites. Our results suggest that coupling between actin and microtubule cytoskeletons driven by Myosin II and KIF20A ensures the spatial coordination between RAB6-positive vesicles fission from Golgi/TGN membranes and their exit along microtubules.

[1] Institut Curie, PSL Research University, CNRS, UMR 144, Molecular Mechanisms of Intracellular Transport, F-75005 Paris, France. [2] Institut Curie, PSL Research University, CNRS, UMR 144, Structural Motility, F-75005 Paris, France. [3] Institut Curie, PSL Research University, CNRS, UMR 144, Cell and Tissue Imaging Facility (PICT-IBiSA), F-75005 Paris, France. [4] ICSN-CNRS, Gif Sur Yvette, F-91190, France. [5] Biokinesis, Paris, F-75008, France. [6] Institut Pasteur, CNRS UMR3691, Membrane Traffic and Cell Division, F-75015 Paris, France. Correspondence and requests for materials should be addressed to S.M.L. (email: stephanie.miserey-lenkei@curie.fr) or to B.G. (email: bruno.goud@curie.fr)

The microtubule (MT) and actin cytoskeletons play important roles in Golgi structure and function. It is now well established that intact MT network and the minus-end MT dynein motor are required for maintaining the Golgi structure. MT depolymerization causes Golgi ribbon fragmentation and Golgi membranes redistribution near endoplasmic reticulum (ER) exit sites. In addition, cis- and trans-Golgi membranes can nucleate and polymerize MTs. This allows the dynamic association of the Golgi complex with the MTOC (MT organizing center) and polarized secretion of Golgi-derived transport carriers toward the leading edge of migrating cells[1]. Actin depolymerization has less pronounced effect on Golgi structure, but leads to its compaction[2]. Several studies have illustrated that actin and actin-binding proteins, including motors of the myosin family, regulate early events of transport biogenesis at the Golgi complex such as protein sorting and membrane fission[3-9].

An important challenge is therefore to better understand how the actin and MT cytoskeletal systems are connected at the Golgi level. Some players have been identified. Connections can be established via the formin family proteins, such as mDia-1, formin-like/FMNL 1 and 2, and INF2 that are present on Golgi membranes[10-13]. A WASP (Wiskott–Aldrich syndrome protein) homolog, WHAMM, directly interacts with MTs and was shown to play a role in the maintenance of

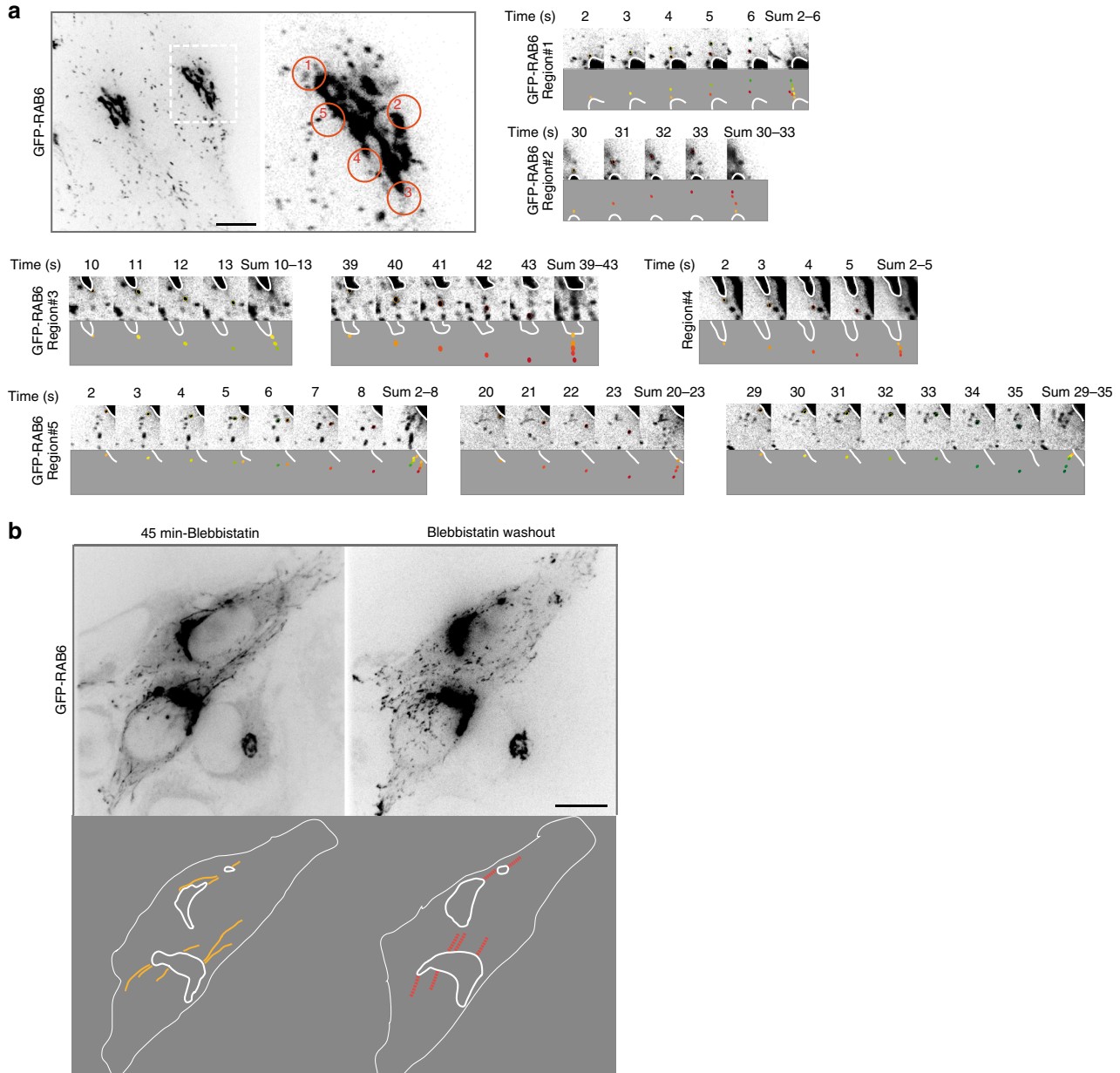

**Fig. 1** GFP-RAB6-positive vesicles exit the Golgi at fission hotspots. **a** HeLa cells stably expressing GFP-RAB6 were imaged every second for 1 min (Supplementary Movie 1). Orange circles show the location of five fission hotspots. Details of each fission event for the five regions of hotspots are displayed. In each case, sequences of images are displayed at the top and a scheme of the events at the bottom. The Golgi region is highlighted by a white line. Vesicles exiting the Golgi are drawn and painted either yellow-green or orange-red. A sum of the events is shown at the end of each sequence. **b** Cells were treated for 45 min with para-nitro-blebbistatin. Then, para-nitro-blebbistatin was washed out. After recovery of normal GFP-RAB6 vesicles trafficking, the cells were imaged every second for 60 s (Supplementary Movie 3). A scheme of the images is displayed at the bottom. The Golgi regions are highlighted by a white line. Tubes are drawn in yellow. Regions where the GFP-RAB6-positive fission hotspots are found are drawn in red. Bars: 10 μm

Golgi structure as well as in ER to Golgi transport[14]. Other candidates are members of the Golgin family, such as the *Drosophila* golgin Lava lamp that interacts both with the dynein/dynactin complex and spectrin[15], and the p230/golgin-245, shown to interact with MACF1, a giant protein that links MTs to the actin cytoskeleton[16].

In this study, we investigated how RAB GTPases, key regulators of intracellular transport and membrane trafficking, and molecular motors control the coupling between actin and

MT cytoskeleton at the Golgi complex. One of the main functions of RAB GTPases is to recruit actin- or MT- based motors on transport carriers, allowing them to move along cytoskeletal tracks. This is the case for RAB6, the most abundant RAB at the Golgi that regulates several transport steps at the Golgi as well as Golgi homeostasis[17–21]. Two closely related RAB6 isoforms, RAB6A and RAB6A′ are present on Golgi/TGN membranes[6,22]. In this manuscript, we will collectively call them RAB6. RAB6 was previously shown to directly or indirectly interact with several

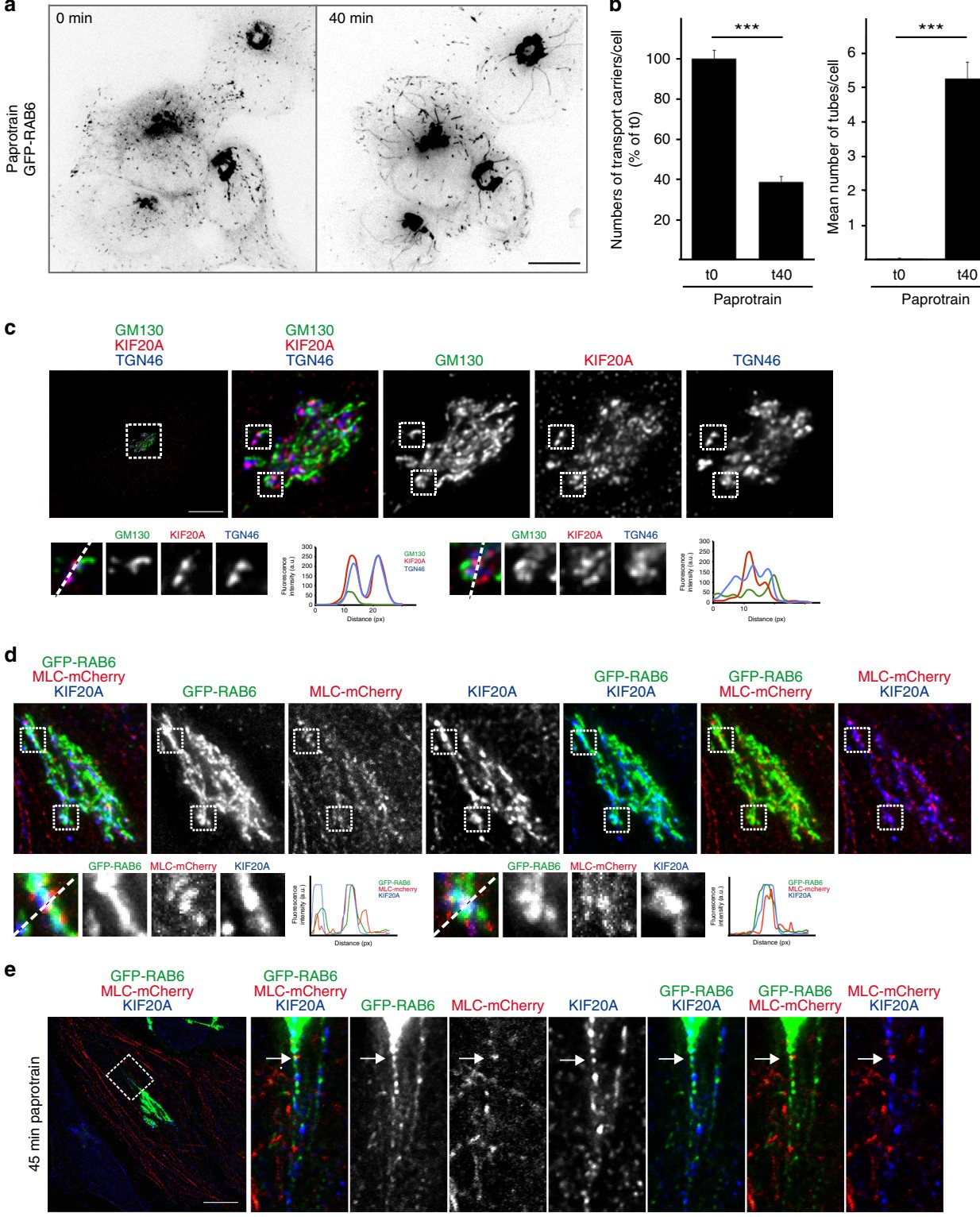

motors, including KIF5B, KIF20A (also known as Rabkinesin-6/MKlp2), the dynein–dynactin complex (via Bicaudal-D), Myosin II and Myosin Va[6,17,23–26]. However, it remains unknown whether RAB6 acts as a platform to couple actin- and MT-associated motors in order to coordinate the function of MTs and actin in Golgi function.

The interaction between RAB6 and Myosin II is critical for the fission of RAB6-positive transport carriers from Golgi/TGN membranes[6]. Here, we show that KIF20A is also involved in the fission process. The coupling between actin and MT cytoskeleton driven by Myosin II and KIF20A ensures the spatial coordination of RAB6-positive vesicles formation at fission hotspots sites and their exit from Golgi/TGN membranes along MTs.

## Results

**RAB6-positive vesicles exit the Golgi complex at fission hotspots**. We have previously shown that RAB6 and Myosin II are implicated in the fission of RAB6-positive transport carriers at the Golgi complex[6]. The inhibition of this process leads to the formation of long membrane tubes connected to the Golgi complex[6]. Detailed analysis of time-lapse microscopy of HeLa cells stably expressing GFP-RAB6 (Supplementary Fig. 1, Supplementary Movie 1) now revealed that RAB6-positive vesicles exit the Golgi complex in defined areas (Fig. 1a, Supplementary Fig. 1 and Supplementary Movie 1). We named them Golgi fission hotspots. A detailed illustration of Golgi fission hotspots for one Golgi is displayed in Fig. 1a. At the optical microscopy resolution, the Golgi fission hotspots are seen at the extremities rather than at the flatter regions of the Golgi. Over 60-s movies, we observed the existence of $6.4 \pm 0.4$ fission hotspots per Golgi ($n = 13$ Golgi) and measured a total of $12.5 \pm 1.3$ vesicles per min exiting the Golgi complex at fission hotspots ($n = 13$ Golgi). This indicates that 1–2 RAB6-positive vesicles exit the Golgi complex at fission hotspots per min (Fig. 1a). The formation of a vesicle usually takes 1 s (Fig. 1a; example in region #1, between time 3 and 4; example in region #5, between time 4 and 5). Movies performed on the same Golgi complex at 10-min interval revealed that the fission hotspots are stable over time (Supplementary Fig. 1, Supplementary Movie 2). If RAB6 vesicles exit the Golgi at preferential sites, one expects that these sites are similar to the ones where the membrane tubes are formed after inhibition of the fission process. To address this point, GFP-RAB6 cells were treated with the Myosin II inhibitor blebbistatin to inhibit fission and to allow the appearance of membrane tubes. Then, blebbistatin was washed out and the Golgi fission hotspots were localized using detailed analysis of time-lapse movies over a 60-s movie. The location of fission hotspots was similar to that of the RAB6-positive membrane tubes obtained after inhibition of the fission process (Fig. 1b, Supplementary Movie 3). Altogether, these results highlight the existence of RAB6-positive Golgi fission hotspots on the Golgi complex.

**KIF20A is involved in the fission process of RAB6-positive vesicles**. To decipher the molecular mechanisms of the fission process of RAB6-positive vesicles from Golgi membranes, we looked for proteins interacting with both Myosin II and RAB6. KIF20A, that we previously identified as a RAB6 effector present on Golgi membranes[25], appeared as an interesting candidate since it was recently shown to directly interact with Myosin II[27,28].

We originally proposed that KIF20A was responsible for the movement of RAB6-positive vesicles along MTs, since dominant negative mutants of KIF20A reduced trafficking[25]. Alternatively, KIF20A might be involved in the generation of RAB6 vesicles. Good evidence later showed that RAB6 vesicles move thanks to KIF 5B[6,17]. Consistent with these results, we found that KIF20A inhibition using paprotrain, a specific chemical inhibitor of KIF20A motor activity[29], did not affect the speed of RAB6-positive vesicles ($0.91 \pm 0.04\,\mu m/s$ in control, $0.89 \pm 0.04\,\mu m/s$ in paprotrain-treated cells, $n = 47$). To address the possible function of KIF20A in the generation of RAB6 vesicles, we performed time-lapse video-microscopy experiments in HeLa cells stably expressing GFP-RAB6 following KIF20A inhibition with paprotrain or its depletion by short interfering RNA (siRNA) (Fig. 2a,b, Supplementary Fig. 2). Interestingly, in both cases, we observed the appearance of long tubules connected to the Golgi complex (Fig. 2a, b, Supplementary Fig. 2, Supplementary Movie 4) and a strong reduction in the number of RAB6-positive vesicles in the cytoplasm (Fig. 2b, Supplementary Fig. 2, Supplementary Movie 4). This phenotype, strikingly similar to the one observed after Myosin II inhibition or its depletion[6], suggests that KIF20A is involved in the fission process of RAB6-positive vesicles from Golgi membranes. Fissions defects were also observed using a second inhibitor BKS0349 which is derived from paprotrain and displays higher affinity (Supplementary Fig. 2) or by over-expressing KIF20 constructs lacking the motor domain (Supplementary Fig. 3). To further confirm that the motor activity of KIF20A is required for the fission of RAB6-positive vesicles, we overexpressed at moderate levels (to avoid bundling of MTs) in control or KIF20A siRNA-treated cells, either wild-type or the K165A mutant of KIF20A (Supplementary Fig. 4). This conserved lysine in the P-loops of kinesins and myosins is absolutely required for nucleotide binding and motor activity[30–32]. As expected, expression of GFP-KIF20A WT rescued the fission defects resulting from KIF20A depletion (Supplementary Fig. 4). In contrast, GFP-KIF20A-K165A expressing cells still displayed membrane tubules connected with the Golgi and a decreased number of vesicles in the cytoplasm (Supplementary Fig. 4). Interestingly, GFP-KIF20A-K165A by itself induced the appearance of Golgi-connected membrane tubules and fission defects, indicating that this construct acts as a dominant negative mutant (Supplementary Fig. 4). Altogether, these results demonstrate that the ATPase activity of KIF20A is required for the fission process of Rab6-positive vesicles.

**Fig. 2** Inhibition of KIF20A function inhibits the fission of Rab6 transport carriers from the Golgi. GFP-RAB6, Myosin II, and KIF20A are co-localized on dotted structures that correspond to the sites of fission. **a** HeLa cells stably expressing GFP-RAB6 were imaged by time-lapse microscopy, and images of the same cell before (t0) or 40 min (t40) after paprotrain (25 μM) treatment are presented (Supplementary Movie 4). Paprotrain was used as an inhibitor of KIF20A function. **b** Quantification of the number of transport carriers and of Golgi-connected tubules in cells treated as indicated in (**a**) (mean ± SEM, $n$ = 38 cells). ***$P < 10^{-10}$ (Student's $t$ test). **c** Staining of endogenous GM130 (green), KIF20A (red), and TGN46 (blue) in HeLa cells indicates a higher co-localization of KIF20A with the *trans*-Golgi marker TGN46 as compared to the *cis*-Golgi marker GM130 (see higher magnifications (boxes) on the bottom). Line profiles of the GM130 (green), the KIF20A (red), and the TGN46 (blue) fluorescence intensities (arbitrary units) along the white-dashed arrow. Bar, 10 μm. **d** Staining of GFP-RAB6 (green), MLC-mCherry (red), and endogenous KIF20A (blue) in HeLa cells indicates a partial co-localization of the three proteins on dotted structures at the Golgi complex (see higher magnifications (boxes) on the bottom). Line profiles of the GFP-RAB6 (green), the MLC-mCherry (red), and the KIF20A (blue) fluorescence intensities (arbitrary units) along the white-dashed arrow. Bar, 10 μm. **e** Staining of GFP-RAB6 (green), MLC-mCherry (red), and endogenous KIF20A (blue) in HeLa cells treated for 45 min with paprotrain indicates a co-localization (arrow) of the three proteins at the base of a GFP-RAB6-positive tube. Bar, 10 μm

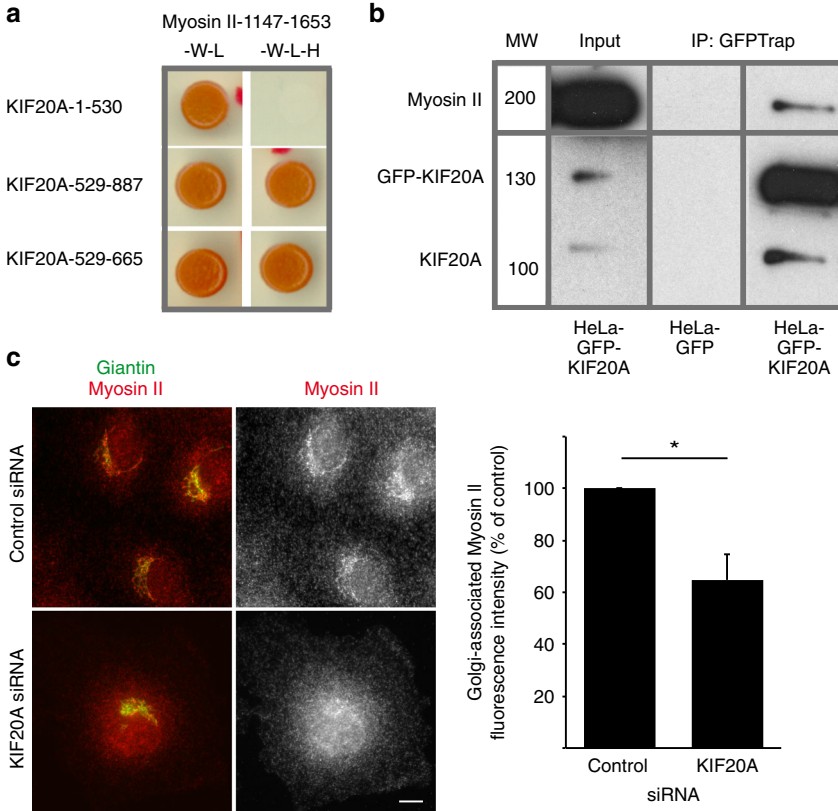

**Fig. 3** KIF20A interacts with Myosin II and stabilizes Myosin II at the Golgi. **a** Yeast two-hybrid interactions between the tail domain of Myosin II (1147–1653 fragment) and different domains of KIF20A. The *Saccharomyces cerevisiae* reporter strain L40 was co-transformed with a plasmid encoding fusion proteins to detect interactions between amino acids 1147 and 1653 of human myosin IIA heavy chain and the motor domain (1–530), the tail domain (529–887), and the RAB6 binding domain (RBD) (529–665) of KIF20A. Growth on medium lacking histidine (-W -L -H) indicates an interaction between the encoded proteins. **b** Endogenous Myosin II is pulled-down by GFP-KIF20A. GFP-KIF20A or GFP-transfected HeLa cell extracts were immunoprecipitated using the GFP-trap system. Myosin II bound to GFP-KIF20A was revealed by western blot analysis using anti-Myosin II antibody. GFP-KIF20A and KIF20A were revealed by anti-KIF20A antibody. Note that endogenous KIF20A is found in the IP fraction because it likely can form dimers with GFP-KIF20A. Input represents a 5% load of the total cell extracts used in all conditions. **c** Staining for endogenous Giantin (green) and endogenous Myosin II using the AD7 antibody (red) in Rat Clone 9 cells 3 days after transfection with specific KIF20A siRNAs. Bar, 10 μm. Of note, KIF20A is required for cytokinesis and KIF20A depletion leads to binucleated cells[56,57]. Quantification of Golgi-associated Myosin II fluorescence intensity in cells treated as described above (mean ± SEM, $n$ = 23–66 cells). *$P$ = 0.02 (Student's $t$ test). MW molecular weight in kDa

Finally, as already reported after RAB6 or Myosin II depletion/inhibition[6], the trafficking of the secretory cargo VSV-G from the Golgi to the plasma membrane was impaired after KIF20A inhibition (Supplementary Fig. 2).

**KIF20A, RAB6, and Myosin II are co-localized at Golgi fission hotspots**. KIF20A can be visualized on fixed cells associated with the Golgi complex[25] (Fig. 2c). The co-localization was more pronounced with the TGN marker TGN46 than with the *cis*-Golgi marker GM130 (Fig. 2c). We validated the Golgi localization of KIF20A using two different antibodies (Supplementary Fig. 5) as well as in a GFP-KIF20A cell line (Supplementary Fig. 5) and we showed that the Golgi signal decreased following KIF20A depletion by siRNA (Supplementary Fig. 5).

We performed a detailed analysis of the extent of co-localization between RAB6, Myosin II, and KIF20A on these Golgi fission hotspots. GFP-RAB6, Myosin II, and KIF20A were found to co-localize on dotted structures (Fig. 2d). On average, 6.3 ± 0.7 of these dots were found per Golgi ($n$ = 21 Golgi). Following KIF20A inhibition with paprotrain, dots containing the three proteins were found at the base of membrane tubes, suggesting that these dots correspond to the sites of fission (Fig. 2e). To further confirm this hypothesis, we treated the cells with blebbistatin to induce the appearance of membrane tubules and then washed out the drug.

As previously shown[6], this leads to membrane fission along the tubules close to membrane bulges that are labeled by RAB6, cargos and F-actin. We observed an accumulation of KIF20A close to the membrane bulges, indicating the presence of KIF20A at the fission sites (Supplementary Fig. 5).

**KIF20A displays two Myosin II binding sites**. KIF20A was recently shown to directly interact with Myosin II[27,28]. The Myosin II binding site has been restricted to the last 60 amino acids of KIF20A and has been shown by co-immunoprecipitation experiments to be the main contributor in Myosin II binding[28]. Using yeast two-hybrid experiments, we confirmed this interaction and identified a second binding site corresponding to the KIF20A-529–665 fragment (Fig. 3a and Supplementary Fig. 6). This fragment was previously characterized as the RAB6 binding domain of KIF20A (RBD)[25]. The tail domain of Myosin II thus interacts with the tail domain of KIF20A via two binding sites (Fig. 3a and Supplementary Fig. 6). However, these two binding sites likely act independently since overexpression of the KIF20A-529-887 domain does not lead to higher fission defects than the overexpression of the KIF20A-529-665 or KIF20A-796-887 domains alone (Supplementary Fig. 3). Consistent with the yeast two-hybrid experiments, GFP-KIF20A and endogenous Myosin II could be immunoprecipitated in the same complexes (Fig. 3b).

**KIF20A recruits and/or stabilizes Myosin II at the Golgi complex**. We then investigated whether KIF20A could be involved in the recruitment and/or stabilization of Myosin II at the Golgi complex. In order to quantify the amount of Myosin II on the Golgi, we used a rat cell line and a Myosin II-specific

antibody, named AD7, whose specificity was previously validated[6,7,33]. KIF20A depletion reduced by 35% the amount of Myosin II associated with the Golgi complex (Fig. 3c). Similar results were obtained using a Myosin light chain specific antibody (Supplementary Fig. 7). Altogether, the above results show that

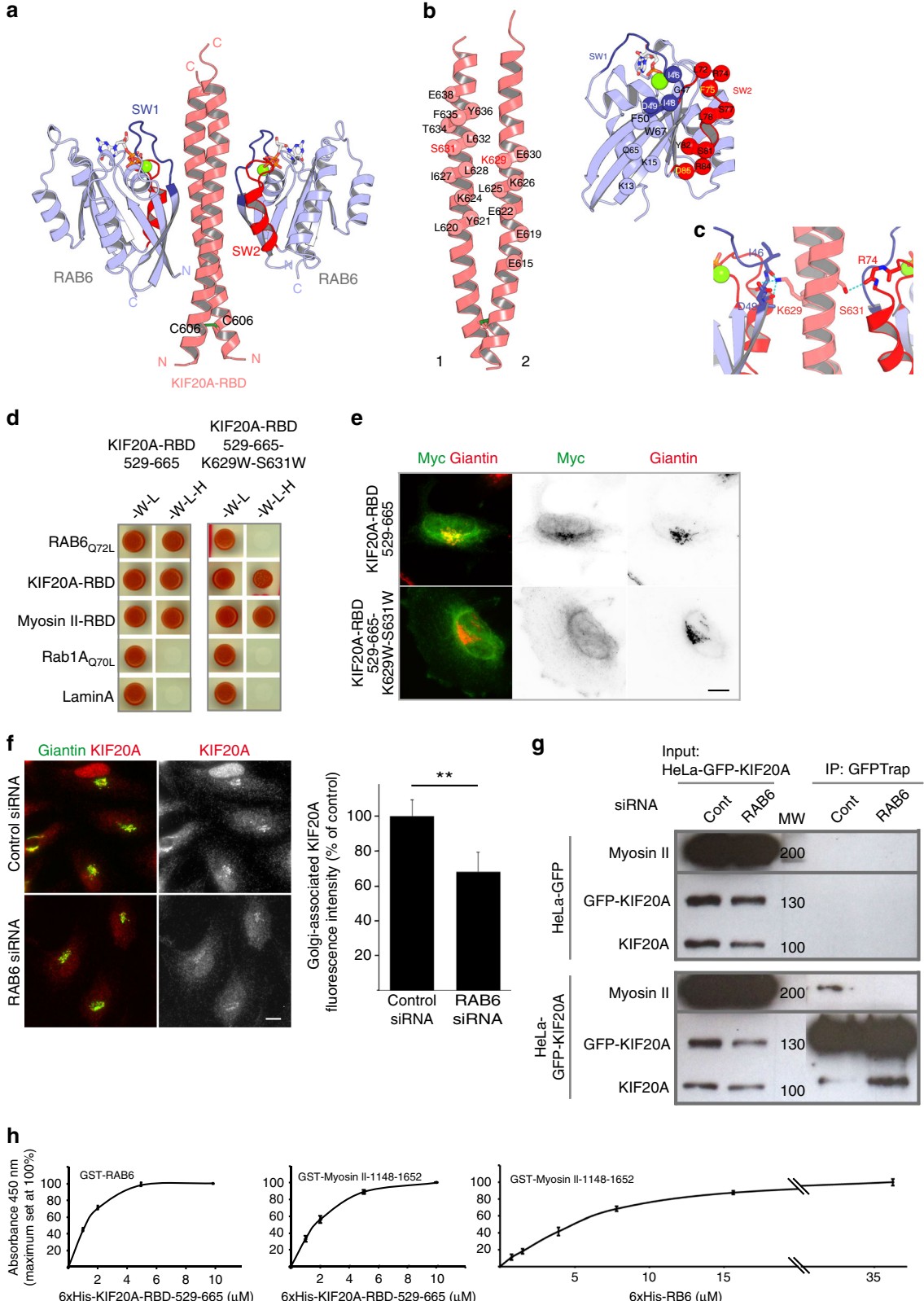

both KIF20A and Myosin II act together in the fission of RAB6-positive vesicles from the Golgi complex.

**RAB6 is required for the Golgi association of KIF20A.** KIF20A interacts directly with RAB6[25]. To further characterize this interaction, we solved the crystal structure of the complex between RAB6 and KIF20A (Fig. 4a–c). The RAB6:KIF20A structure reveals that the 603–645 residues of KIF20A represent the RBD. KIF20A-RBD is a dimer composed of two parallel alpha helices that form a right-handed coiled-coil additionally stabilized by an inter-helical cysteine bridge (Fig. 4a). Both RBD helices participate in RAB6 binding, two RAB6 molecules binding at opposite sides of the dimer (Fig. 4a–c). The RAB6:KIF20A interface is formed by complementary surfaces of the partners (Fig. 4b) that mediate hydrophobic and polar contacts between the molecules. KIF20A K629 and S630 residues are located in the central part of the RAB6-binding site (Fig. 4b, c) and form polar contacts with the partner (Fig. 4c). Prediction based on the structure indicates that substitution of these polar residues by bulky hydrophobic residues should substantially alter the RAB6-binding site. Indeed, the KIF20A (529–665) fragment with K629W and S630W mutations lost its capacity to interact with RAB6 Q72L by yeast two-hybrid (Fig. 4d). We have previously shown that overexpressed KIF20A-RBD can be recruited to the Golgi complex[25]. A myc-tagged version of the KIF20A-529-665-K629W-S630W fragment overexpressed in HeLa cells remained mostly cytosolic in contrast to the Golgi-localized wild-type form (Fig. 4e), demonstrating the importance of the RAB6:KIF20A direct interaction for KIF20A recruitment to Golgi. The mutations do not affect KIF20A-RBD dimerization and importantly its ability to interact with the tail domain of Myosin II (Fig. 4d). This suggests that RAB6 and Myosin II may bind to the KIF20A dimeric coiled-coil (603–665) in different ways. Since this site has two RAB6-binding sites, one of them might be used for Myosin II binding rather than for RAB6 binding and the tri-partite complexes between KIF20A, RAB6, and Myosin II can still be formed. Finally, depletion of RAB6 reduced by 40%, the amount of KIF20A associated with the Golgi complex (Fig. 4f), indicating that RAB6 contributes to the recruitment and/or the stabilization of KIF20A at the Golgi. Since the recruitment of KIF20A on the Golgi only partially depends on RAB6, it is likely that additional mechanisms are implicated or alternatively that the Golgi-associated pool of KIF20A is very stable.

To further understand this process, we tested whether RAB6 is required for KIF20A/Myosin II interaction. Co-immunoprecipitation experiments showed that the complex between KIF20A and Myosin II was lost in RAB6-depleted cells (Fig. 4g). Similar results were obtained in MEF cells derived from RAB6 conditional KO mice embryos and incubated in vitro with tamoxifen to deplete RAB6 (Supplementary Fig. 7).

We next measured the relative binding affinities of RAB6 for Myosin II, RAB6 for KIF20A, and KIF20A for Myosin II using a previously described solid-phase assay[34]. These experiments revealed that RAB6 and Myosin II-1148-1652 bind to KIF20A-RBD-529-665 with similar affinities in the micromolar range (Fig. 4h). On the other hand, RAB6 binds to Myosin II-1148-1652 with around five times lower affinity (Fig. 4h). The fact that RAB6 and Myosin II display the highest affinities for KIF20A suggests that KIF20A is central for the interaction between RAB6 and Myosin II. A likely scenario is that RAB6 first recruits KIF20A, which then binds to Myosin II (see model on Fig. 6c and discussion).

**KIF20A limits RAB6 diffusion at the Golgi ensuring RAB6 localization at fission hotspots.** At the metaphase/anaphase transition, KIF20A was shown to stabilize the central spindle thanks to its additional C-terminal MT-binding domain and to promote the localization on MTs of several proteins, including Plk1 and Aurora B[35–37]. A tempting hypothesis is that KIF20A fulfils a similar role on Golgi membranes to anchor RAB6 near MTs, which would favor the loading of RAB6 vesicles onto MTs.

To test this hypothesis, we performed FRAP (fluorescence recovery after photobleaching) experiments in HeLa cells stably expressing GFP-RAB6 (Fig. 5a, b, Supplementary Movie 5). After FRAP on Golgi extremities, a wave of RAB6 fluorescence coming from the core of the bleach area to the extremity of the Golgi was observed (Fig. 5b, Supplementary Movie 5). Following KIF20A inhibition with paprotrain, the wave of diffusion corresponding to RAB6 fluorescence was faster (Fig. 5c, Supplementary Movie 5) as exemplified at time 5 s. The half-time recovery after photobleaching was also reduced by 30% as compared to control cells (Fig. 5d, e), reflecting a higher mobility of GFP-RAB6 on Golgi membranes (Fig. 5d, e).

The above results indicate that KIF20A limits the diffusion of RAB6 molecules on Golgi membranes and confine them into membrane areas close to MTs.

**KIF20A co-localizes with Golgi-associated growing MTs on Golgi membranes.** After fission from Golgi membranes, RAB6-positive vesicles are transported along MTs to the cell periphery[17]. Golgi membranes can nucleate and polymerize MTs that serve in particular to sustain polarized secretion[38–40]. We thus

**Fig. 4** Definition of the RAB6-BD of KIF20A. RAB6 partially recruits KIF20A on the Golgi complex. RAB6 is required for KIF20A-Myosin II interaction. Relative binding affinities between RAB6, KIF20A, and Myosin II. **a–c** KIF20A-RAB6-binding domain (KIF20A-RBD) is a dimer composed of parallel helices that form a right-handed coiled-coil stabilized by an inter-helical cysteine bridge and two RAB6 molecules bind on opposite sides of the dimer **b**. The RAB6:KIF20A interface makes hydrophobic and polar contacts between the molecules **b**, **c**, in particular K629 and S631, are buried in the interface and involved in direct interactions at the center of the interface. **d** Yeast two-hybrid interactions between the KIF20A-RBD-529-665 fragment or KIF20A-RBD-529-665-K626W-S631W fragment with RAB6, Myosin II-RBD, and KIF20A-RBD. The *Saccharomyces cerevisiae* reporter strain L40 was co-transformed with a plasmid encoding fusion proteins to detect interactions between the KIF20A-RBD-529-665 fragment or KIF20A-RBD-529-665-K626W-S631W fragment and RAB6-Q72L, KIF20A-RBD, Myosin II-RBD, RAB1Q70L, and Lamin A. Growth on medium lacking histidine (-W -L -H) indicates an interaction between the encoded proteins. **e** Golgi association of the KIF20A-RBD-529-665 fragment or KIF20A-RBD-529-665-K626W-S631W fragments. Staining for endogenous Giantin (red) and overexpressed myc-tagged KIF20A-RBD-529-665 fragment or KIF20A-RBD-529-665-K626W-S631W fragment. **f** Staining of endogenous Giantin (green) and KIF20A (red) in HeLa cells 3 days after transfection with specific RAB6 siRNAs. Quantification of Golgi-associated KIF20A fluorescence intensity in cells treated as described above (mean ± SEM, n = 34-37 cells). **P < 10⁻³ (Student's *t* test). Bar, 10 μm. **g** RAB6 is required for KIF20A/Myosin II interaction. GFP-KIF20A or GFP expressing HeLa cell extracts treated for 3 days with control or RAB6-specific siRNAs were immunoprecipitated using the GFP-trap system. Myosin II bound to GFP-KIF20A was revealed by western blot analysis using anti-Myosin II antibody. Input represents a 5% load of the total cell extracts used in all conditions. **h** 96-well plates were coated with recombinant GST-RAB6 or GST-Myosin II-1148-1652 and incubated with increasing amounts of recombinant 6 × His-KIF20A-RBD-529-665 or 6 × His-RAB6, as indicated (solid-phase assay). After washes, KIF20A-RBD-529-665 or RAB6 proteins bound to RAB6 or Myosin II-1148-1652 were detected (arbitrary units) using anti-6 × His antibodies and a chromogenic substrate (mean ± SEM, n = 3 experiments). No binding to GST-GFP alone was detected. MW molecular weight in kDa

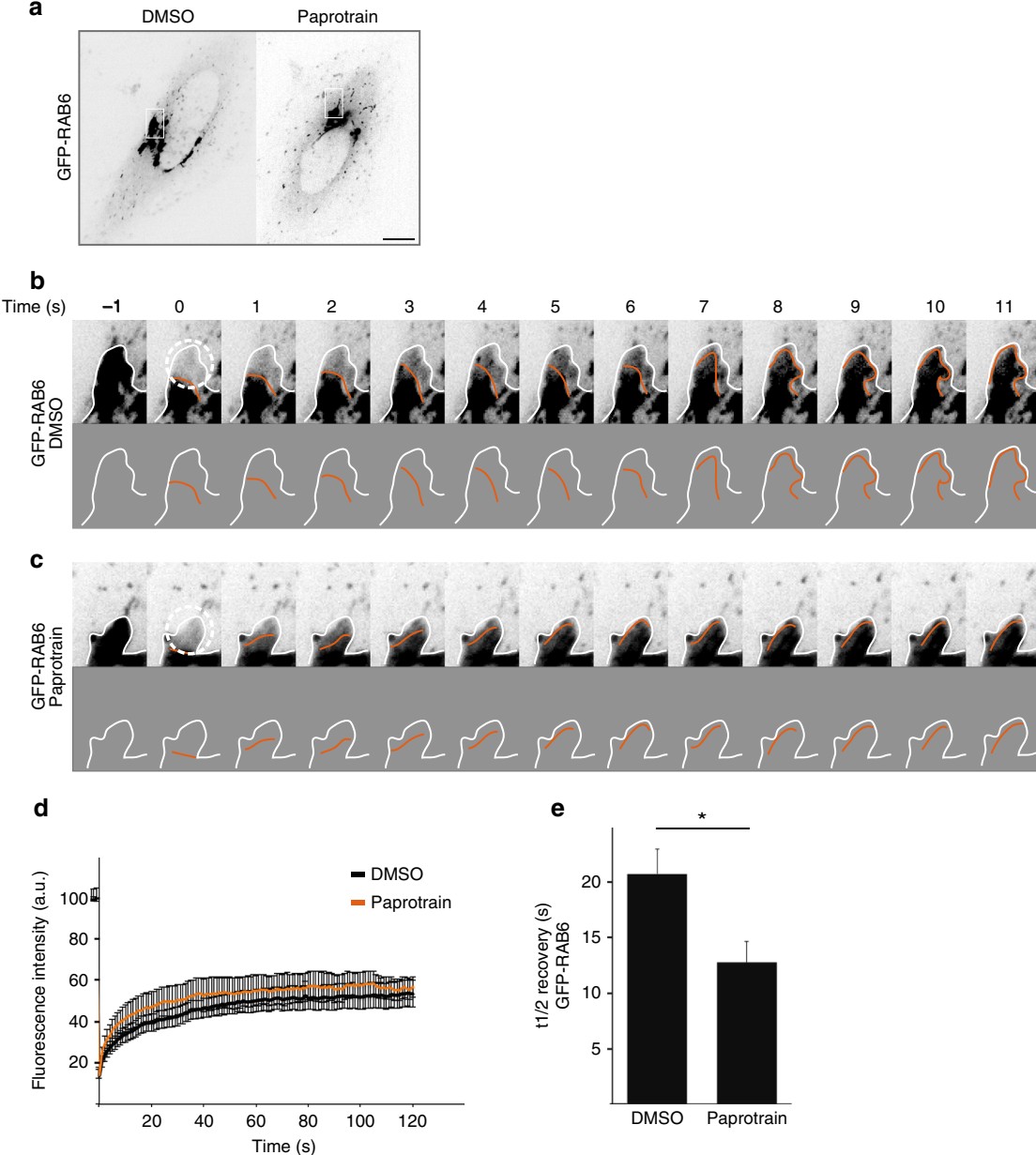

**Fig. 5** GFP-RAB6 diffuses on Golgi membranes and KIF20A inhibition affects the diffusion of GFP-RAB6. **a** Images of GFP-RAB6 expressing cells treated with DMSO or with paprotrain for 30 min. HeLa cells expressing GFP-RAB6 were bleached in a 30 pixels circular region of the Golgi apparatus (white circle) and then imaged for 2 min by spinning disk microscopy. Sequence of images of GFP-RAB6 in Golgi of a control cell (**b**) and a paprotrain-treated cell (**c**) before and after photobleaching. The round region delimited by a dotted line was photobleached immediately after the first image (−1). In both cases, images are displayed at the top and schematics of the images at the bottom. The orange line represents the recovery of the GFP-RAB6 fluorescence. In **b**, the photobleached area rapidly recovered fluorescence. The fluorescence flows from the center of the Golgi to the extremity of the bleached area. In **c**, the photobleached area recovered fluorescence twice faster than in **b** (compare t5). **d** The fluorescence recovery after photobleaching was measured in three independent experiments and plotted for GFP-RAB6 treated with DMSO (black) or with paprotrain (orange) (n = 17–19 Golgi). **e** Half-life recovery of GFP-RAB6 fluorescence in control cells and cells treated with paprotrain (mean ± SEM, n = 17–19 cells). *P = 0.01 (Student's t test)

wondered if KIF20A interacts with MTs that polymerize on Golgi membranes. First, we observed that after nocodazole treatment and washout, tubulin dots corresponding to reforming MTs co-localize with KIF20A (Fig. 6a). Second, Myosin II and RAB6 are found co-localized on peripheral dots with GCC185, a TGN Golgin known to recruit the MT + TIP CLASP proteins to the TGN, allowing the nucleation and formation of Golgi MTs[38] (Fig. 6b). Quantification indicated that there are 7.51 ± 0.46 dots of triple co-localization per Golgi (n = 25 Golgi). Altogether, these results indicate that RAB6, KIF20A, and Myosin II are

co-localized on dots corresponding to places of polymerization of MTs originating from Golgi membranes.

## Discussion

The main finding of this study is the existence of preferential sites for the fission of RAB6-positive post-Golgi secretory vesicles on Golgi/TGN membranes. Hotspots for the fission of endocytic vesicles and ER-derived vesicles (ER-exit sites) as well as exocytic hotspots[41] have been previously described at the plasma or ER

membranes, but to our knowledge never on Golgi/TGN membranes.

We measured around six hotspots per Golgi. Remarkably, this number corresponds to the number of spots where RAB6, Myosin II, and KIF20A co-localize and to the number of

membrane tubes observed after the inhibition of Myosin II (n = 5, ref. [6]), that of KIF20A (n = 5, Fig. 2) or RAB6 depletion (n = 4, ref. [6]). This suggests that a connection between actin and MT cytoskeletons established by an actin-based motor and a MT-based motor that interact with each other is necessary for the

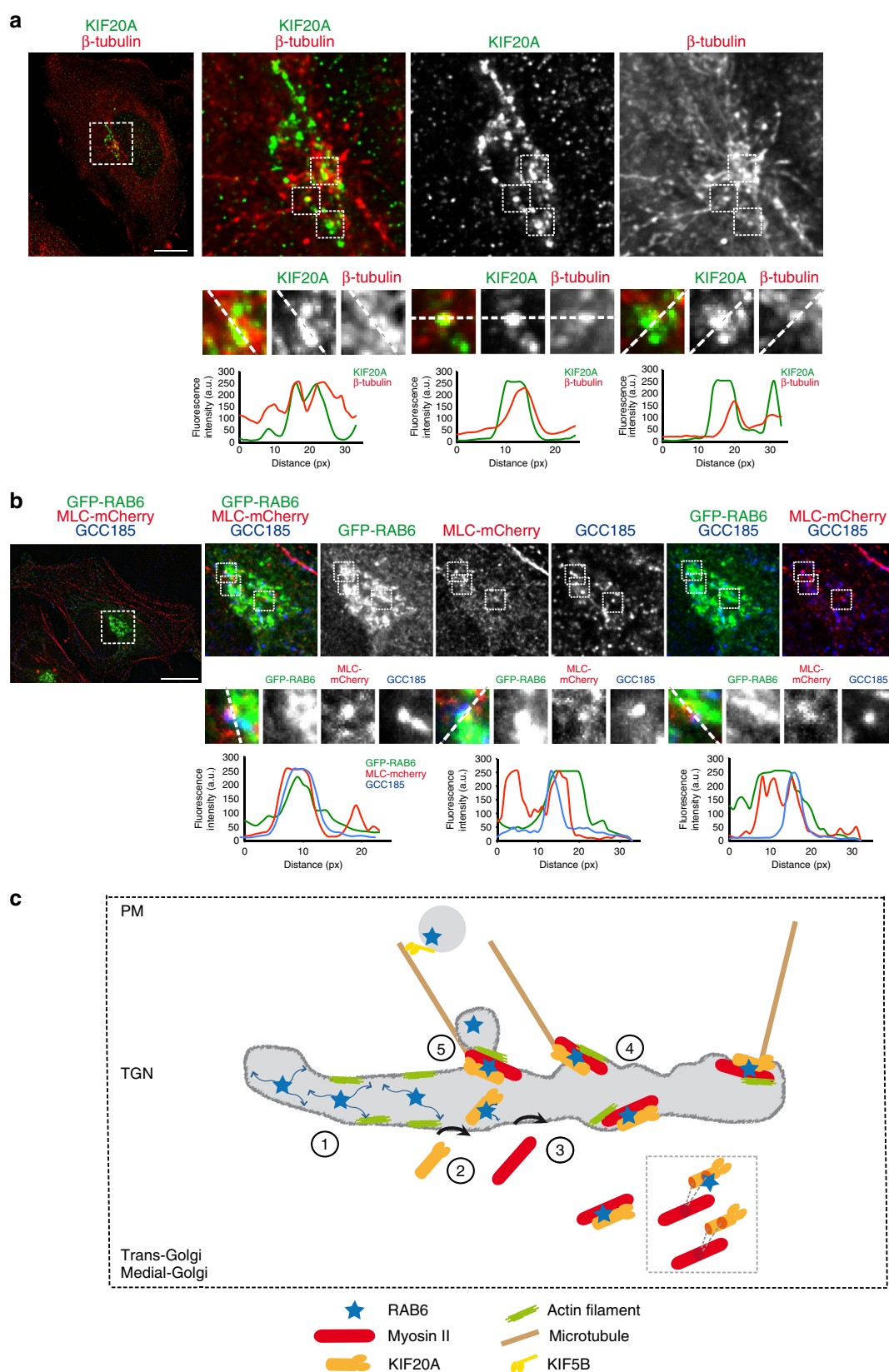

generation of Golgi fission hotspots. It is likely however that additional mechanisms determine where and when hotspots can form. A tentative hypothesis is that some RAB6 molecules are concentrated in domains, maybe enriched in specific lipids such as PtdIns-4-phosphate. The existence of such domains has been postulated[42] but not yet demonstrated. In addition, other fission promoting molecular complexes such as CtBP1-S/BARS, members of the 14-3-3 family, phosphoinositide kinase PI4KIIIβ, and lysophosphatidic acid acyltransferase δ have been identified[43–45]. It will be important to test in future experiments whether KIF20A/RAB6/Myosin II complexes are physically or functionally related to these complexes. Another important issue will be to investigate the relationships between the budding and fission machineries of RAB6-positive vesicles. RAB6 depletion leads to the accumulation of budding profiles on Golgi membranes[46] that are coated with clathrin and COPI[18]. This suggests that the fission machinery driven by Myosin II and KIF20A is distinct from that required for budding.

The sequence of events that can be envisioned for the generation of fission hotspots is the following (Fig. 6c). RAB6 depletion leads to a decrease of Golgi-associated KIF20A, indicating that RAB6 participates, at least in part, to KIF20A recruitment on Golgi/TGN membranes. This would allow the clustering of RAB6 molecules and surrounding membranes to MTs. In support of this hypothesis, we showed that KIF20A inhibition increases the diffusion rate of RAB6 on Golgi/TGN membranes. Such a role for KIF20A is reminiscent of the one proposed during mitosis where KIF20A promotes the MT-dependent localization of Plk1, Aurora B, and Cdc14A to the central spindle[35]. Myosin II would then be recruited by KIF20A and RAB6 and would drive the fission of RAB6-positive transport carriers from Golgi/TGN membranes. An alternative scenario is that Myosin II is first recruited and then interacts with KIF20A. This is however unlikely since depletion of KIF20A decreases the amount of Myosin II associated with Golgi/TGN membranes.

RAB6-positive transport vesicles move along MTs toward the cell periphery and the plasma membrane to convey secretory cargoes such as VSV-G or NPY[6,17]. Whether the MTs used originate from MTOC or Golgi membranes is unknown, but our observations suggest that at least part of these MTs could originate from Golgi membranes. Indeed, KIF20A co-localizes with MTs that regrow from Golgi membranes. In addition, GCC185, a golgin involved in the formation of Golgi-associated MTs, could also be visualized on structures containing RAB6 and Myosin II. Importantly, the motor activity of KIF20A appears to be required for a normal fission process (Supplementary Figs. 2–4). As KIF20A is not the kinesin moving RAB6-positive vesicles along MTs (KIF5B is responsible for this, see refs. [6,17]), two possibilities can be envisioned: the motor activity of KIF20A could be important to generate a level of membrane tension

required for vesicle fission or alternatively it could be essential for the local organization of MTs to favor the right positioning of the fission machinery.

We previously showed that RAB6 controls the fission of its own transport carriers, which provides a mechanism for coordinating several events involved in the biogenesis of transport vesicles. Our study illustrates that in addition RAB6 could select a subset of MTs for transport of its own transport carriers. It will be interesting to see whether this is a unique mechanism or if other RAB GTPases that interact with actin and MT-based motors could function in the same way for the biogenesis of transport vesicles.

## Methods

**Antibodies and reagents.** The following reagents were used: Para-Nitroblebbistatin (OPTOPHARMA Ltd), Y-27632 (Calbiochem), ML7 (Calbiochem), Paprotrain (Enzo Life Sciences). The following antibodies were used for biochemical experiments: Mouse anti-GM130 (BD Biosciences, #610822), sheep anti-TGN46 (AbD Serotec, AHP500), rabbit anti-KIF20A (ref. [25]; Bethyl Laboratory, #A300-879A; proteintech, #15911-1-AP), goat anti-KIF20A (Santa Cruz Biotechnology, sc-104954), rabbit anti-RAB6 (ref. [47] and Santa Cruz Biotechnology, sc-310), human anti-Giantin (Recombinant antibody platform, Institut Curie, Paris, France); rabbit anti-myc (Santa Cruz Biotechnology, sc-789), rabbit anti-MHC (Myosin II Heavy Chain) (Bethyl Laboratory, A304-490A), human anti β-tubulin (Recombinant antibody platform, Institut Curie), rabbit anti-GCC185 (P. Gleeson, University of Melbourne, Australia), mouse anti-actin (Sigma, A-2228), rabbit ant-Cul3 (Bethyl Laboratory, A-301-110A), mouse anti-poly-histidine antibody (Recombinant antibody platform, Institut Curie). The plasmids encoding the following fusion proteins were used: GFP-RAB6 (F. Perez, Institut Curie), mCherry-RAB6, MLC-mCherry (A.M. Lennon-Dumenil, Institut Curie), GFP-KIF20A-529-665[25], GFP-KIF20A-529-887[25], GFP-KIF20A-25-887 (I.-M. Yu, Institut Curie).

**Cell culture and transfection.** HeLa[25] and Rat Clone 9 cells (ATCC #CRL1439) were grown in DMEM medium (Gibco BRL) supplemented with 10% fetal bovine serum, 100 U/ml penicillin/streptomycin, and 2 mM glutamine. Both cell lines were tested free for mycoplasma contamination. Cells were seeded onto 6-well plates, 18-, or 12-mm glass coverslips or Fluorodish and grown for 24 h before transfection. HeLa cells stably expressing GFP-RAB6′ or GFP-KIF20A (KIF20A-BAC, a kind gift from A. Hyman, Dresden, Germany) were cultivated in the presence of 50 μg/ml geneticin (Gibco BRL). For expression of the constructs used in this study, HeLa cells were transfected using either the calcium phosphate precipitation method or X-tremeGENE9 (Roche) following the manufacturer's instructions. For silencing experiments, HeLa and Rat Clone 9 cells were transfected with the corresponding siRNA once (in the case of RAB6A/A′ siRNAs) or twice at 24 h interval (in the case of KIF20A siRNAs) using HiPerFect (Qiagen) following the manufacturer's instructions. For rescue experiments, cells were transfected with GFP-KIF20A or GFP-KIF20A-K165A 24 h after transfection with KIF20A siRNAs. Forty-eight hours after this second round of transfection, cells were proceeded for video-microscopy experiments. The N-terminal 25 amino acids truncated version of KIF20A was used because of better expression efficiency.

**RNA interference.** The sequences of the siRNAs used in this study are the following: human RAB6A/A′: (GACAUCUUUGAUCACCAGA)[48]. These oligonucleotides were obtained from Sigma (Paris, France). For silencing human KIF20A, different sequences were used: one described in ref. [36] (obtained from Sigma) and specific SMART pools chemically synthesized by Dharmacon Research,

**Fig. 6** KIF20A co-localizes with growing microtubules on Golgi membranes. **a** HeLa cells were incubated for 45 min at 4 °C to promote microtubule depolymerization (without affecting the Golgi morphology) and then incubated for 3 min at RT to allow microtubule repolymerization. Staining of endogenous KIF20A (green) and β-tubulin (red) in HeLa cells indicates a partial co-localization of KIF20A with growing microtubules from Golgi membranes (see higher magnifications (boxes) on the bottom). Line profiles of the KIF20A (green) and the β-tubulin (red) fluorescence intensities (arbitrary units) along the white dashed line. Bar, 10 μm. **b** Staining of GFP-RAB6 (green), MLC-mCherry (red), and endogenous GCC185 (blue) in HeLa cells indicates a partial co-localization of the three proteins on dotted structures at the Golgi complex (see higher magnifications (boxes) on the bottom). Line profiles of the GFP-RAB6 (green), the MLC-mCherry (red), and the GCC185 (blue) fluorescence intensities (arbitrary units) along the white dashed line. Bar, 10 μm. **c** Schematic of the sequence of events that can be envisioned for the generation of fission hotspots at the TGN. (1) RAB6 diffuses on Golgi/TGN membranes. (2) RAB6 participates in the recruitment and stabilization of KIF20A. When bound to KIF20A, RAB6 diffusion is decreased allowing the localization and anchoring of RAB6 molecules to sites of growing Golgi-associated microtubules. (3) Myosin II is recruited by KIF20A and RAB6. Either a complex between RAB6 and KIF20A is required for Myosin II recruitment, or KIF20A acts alone. Myosin II can be recruited by the KIF20A-RBD-529-665 domain in complex or not with RAB6 or through the 796–887 domain. (4) KIF20A, RAB6, and Myosin II, in association with actin filaments and microtubules, define a Golgi fission hotspot. There are around six hotspots per Golgi. (5) Myosin II and actin then drive the fission of RAB6-positive transport carriers from Golgi/TGN membranes. RAB6-positive vesicles are then transported along microtubules to the plasma membrane thanks to KIF5B

Inc. SiRNA targeting luciferase (CGUACGCGGAAUACUUCGA) was used as a control and was obtained from Sigma.

**Yeast two-hybrid experiments**. A yeast two-hybrid screen of a human placental cDNA library with mouse KIF20A-529-887 as a bait was performed by Hybrigenics SA (www.hybrigenics.com). Several clones corresponding to human non-muscle Myosin II heavy chain gene have been isolated. These clones correspond to nucleotides 3984–4413 of the CDS of mouse Myosin IIA. Different sequences corresponding to nucleotides 3744–4122, 3744–3984, 3987–4413 of the CDS of human Myosin IIA have been amplified by PCR and inserted into the SmaI site of yeast two-hybrid pGADGH vector. Yeast two-hybrid experiments were performed as described in ref. [6] except that the LexA fusion proteins correspond to different domains of KIF20A described in ref. [49].

**Construction of KIF20A mutants**. For KIF20A-529-665-K629W-S631W mutants: single-point mutations were inserted in the mouse KIF20A sequence at position 1884, 1887, and 1893 using the QuickChange mutagenesis kit (Agilent). For mCherry-KIF20A-796-887: the 2388-2661 fragment of mouse KIF20A was amplified by PCR and inserted into a mCherry vector allowing the mammalian and bacterial expression of mCherry-KIF20A-796-887. For GFP-KIF20A-25-665: the 75-1995 fragment of human KIF20A was amplified by PCR and inserted into a pOPIN-GFP vector (Addgene) allowing the mammalian expression of GFP-KIF20A-25-665. For GFP-KIF20A-25-887-K165A: single-point mutation was inserted in the mouse KIF20A sequence at position 495 using the QuickChange mutagenesis kit (Agilent).

**Co-immunoprecipitation and western blot experiments**. To test the interaction between GFP-KIF20A and endogenous Myosin II, HeLa cells transfected or not with RAB6 siRNA were trypsinized, washed once in PBS, and incubated on ice for 60 min in a lysis buffer: 25 mM Tris pH 7.5, 50–100 or 200 mM NaCl, and 0.1% NP40. Cells were then centrifuged 10 min at 10,000×g to remove cell debris. Extracts were then processed for co-immunoprecipitation using GFP-trap (Chromotek) following manufacturer's instructions.

To test the interaction between RAB6 and endogenous Myosin II and KIF20A, MEF cells treated or not with Tamoxifen for 96 h (for details see ref. [46]) were trypsinized, washed once in PBS, and incubated on ice for 60 min in a lysis buffer: 25 mM Tris pH 7.5, 50–100 or 200 mM NaCl, 0.1% NP40, protease inhibitor cocktail (Sigma). Cells were then centrifuged 10 min at 10,000×g to remove cell debris. Extracts were then processed for co-immunoprecipitation using 2 μg of anti-KIF20A antibody coupled to Protein G-Sepharose beads for 4 h at 4 °C in lysis buffer. A Rabbit anti-Culin3 antibody was used as control IgG. Beads were washed four times in lysis and then processed for western-blotting.

For western-blotting experiments, cells were processed as in ref. [6]. Cell solubilization was performed in 25 mM Tris pH 7.5, 50 mM NaCl, 0.1% NP40, and a protease inhibitor cocktail (Sigma). The following primary antibodies were used: rabbit anti-MHC (Covance; 1:2000), rabbit anti-RAB6 (Santa Cruz 1:1000 or ref. [47] 1:2000), mouse anti-GFP (Roche; 1:1000), rabbit anti-KIF20A (Bethyl or A174[25]; 1:1000). Secondary Horseradish Peroxidase (HRP)-coupled antibodies were from Jackson Laboratories.

**Immunofluorescence microscopy**. HeLa or Rat Clone 9 cells grown on coverslips were fixed either in 4% paraformaldehyde (PFA) for 15 min at RT, in methanol (2 min, −20 °C) or in TCA (10% TCA 20 min at 4 °C, followed by a 3 min permeabilization in PBS, 0.1% Triton X-100). For mAD7 staining, cells were treated for 2 min with 0.1% Triton X-100 after fixation with PFA. Cells were then processed for immunofluorescence as previously described[48]. The following primary antibodies were used: human anti-Giantin (1:200, ref. [50]), mouse anti-GM130 (1:1000), mouse AD7 anti-myosin II (1:50 to 1:200, ref. [33]), rabbit anti-GCC185 (1:100), rabbit anti-KIF20A (1:2000 to 1:8000, ref. [25]), goat anti-KIF20A (1:2000 to 1:8000), rabbit anti-MLC (1:200, Cell Signaling), human and rabbit anti-β tubulin (1:200, ref. [50]), and sheep anti-TGN46 (1:1000). Fluorescently coupled secondary antibodies were obtained from Jackson. Coverslips were mounted in Mowiol and examined under a three-dimensional deconvolution microscope (Leica DM-RXA2), equipped with a piezo z-drive (Physik Instrument) and a 100 × 1.4NA-PL-APO objective lens for optical sectioning or a DMRA Leica microscope with a 63× objective lens. Three-dimensional or one-dimensional multicolor image stacks were acquired using the Metamorph software (MDS) through a cooled CCD camera (Photometrics Coolsnap HQ). For deconvolution, cell images were acquired as described in ref. [51].

**Time-lapse fluorescence microscopy**. Transfected cells were grown either on glass bottom Fluorodish or on glass coverslips and transferred, just before observation, to custom-built aluminium microscope slide chambers (Ludin chamber, LIS). Time-lapse imaging was performed at 37 °C using a spinning-disk microscope mounted on an inverted motorized microscope (Nikon TE2000-U) through a 100 × 1.4NA PL-APO objective lens. The apparatus is composed of a Yokogawa CSU-22 spinning-disk head, a Roper Scientific laser launch, a Photometrics Coolsnap HQ2 CCD camera for image acquisition and Metamorph software (MDS) to control the setup. Acquisition parameters were 100 ms exposure for GFP

channel and 100 ms for mCherry channel. Laser was set to 30% in each case. Images shown in figures correspond to the maximal intensity projection through the Z axis performed with the Image J software (NIH Image). For FRAP analyses, HeLa cells stably expressing GFP-RAB6 were maintained in culture medium in glass bottom Fluorodish cell culture dishes and imaged on a similar spinning disk microscope equipped with FRAP head (Errol and Roper). Images were collected before bleaching of a rounded 30 pixels region of the Golgi apparatus every second for 5 s and after photobleaching every 1 s for 2–3 min. Images were processed using the Metamorph software. After correction of the photo-bleaching due to acquisition, the background was subtracted. The intensity of fluorescence was then normalized and plotted on the graph.

**Quantification of the number of transport carriers and membrane tubes**. A total of 16–50 transfected cells were imaged for each condition. When cells were treated with drugs, quantifications were performed on the same cells, before and after drug treatment. The maximal intensity projection through the z axis of each image stack was done using the Image J software. The number of transport carriers (corresponding to transport carriers moving at least on three following snapshot) and tubes (connected to the Golgi complex) was manually counted using the cell counter macro from ImageJ (NIH Image).

**Quantification of Myosin II and KIF20A Golgi-associated fluorescence intensity**. Images of control or depleted cells were acquired using the same parameters, without automatic scaling and gain adjustment, avoiding saturated pixels. In all cases, the Golgi area was defined by the Giantin labeling, and the absolute intensity of Golgi-associated Myosin II or KIF20A in this area was measured using Image J software (NIH Image). In the case of KIF20A depleted cells, since cells are multinucleated and have a bigger Golgi, the KIF20A Golgi-associated fluorescence intensity was normalized to the size of the Golgi.

**Recombinant protein purification and solid-phase assays**. GST fused to Myosin II-1148-1652 was expressed in the BL21 (DE3) strain of *Escherichia coli* after induction with 500 μM isopropyl-β-d-thiogalactopyranoside at 20 °C overnight. Cells were lysed in PBS, 2 mM β-mercaptoethanol, protease inhibitor and lysozyme by cell disruption. The GST fusion proteins were affinity-purified using Gluthatione Sepharose 4B (GE Healthcare) and eluted with 50 mM Tris pH 8, 300 mM NaCl, 2 mM β-mercaptoethanol, 5% glycerol, and 10 mM reduced glutathione. Purified proteins were then dialyzed against a solution containing 20 mM Hepes pH 7.5, 100 mM NaCl, 2 mM MgCl2. GST fused to wild-type RAB6A was expressed in the BL21 codon plus (DE3)-RIL strain of *E. coli* after induction with 400 μM isopropyl-β-d-thiogalactopyranoside at 20 °C overnight. Cells were lysed by cell disruption in 50 mM Hepes pH 7.5, 500 mM LiCl, 2 mM β-mercaptoethanol, 10 μM GTP, protease inhibitor, PMSF, and lysozyme. The GST fusion proteins were affinity-purified using gluthatione sepharose 4B (GE Healthcare) and eluted with 50 mM Hepes pH 7.5, 500 mM LiCl, 2 mM β-mercaptoethanol, 10 μM GTP, and 15 mM reduced glutathione. Purified proteins were then dialyzed against a 2 L solution containing 25 mM Hepes pH 7.5, 50 mM NaCl, 2 mM β-mercaptoethanol, 2 mM MgCl2, 10 μM GTP. 6xHis-wild-type RAB6A was expressed in the BL21 codon plus (DE3)-RIL of *E. coli* after induction with 400 μM isopropyl-β-d-thiogalactopyranoside at 30 °C for 4 h. Cells were lysed by cell disruption in 50 mM HEPES pH 7,5, 500 mM LiCl, 2 mM β-mercaptoethanol, 1 mM MgCl2, 10 μM GTP, protease inhibitors, PMSF, and lysozyme. Proteins were affinity-purified using HiTrap™ Chelating HP (GE Healthcare) and eluted with 50mM Hepes pH 7,5, 500 mM LiCl, 2 mM β-mercaptoethanol, 1mM MgCl2, 10 μM GTP, 500 mM Imidazole. His-tags were removed by recombinant Tobacco Etch Virus (rTEV) protease cleavage during overnight dialysis in 50 mM Hepes pH 7,5, 100 mM NaCl, 2 mM β-mercaptoethanol, 1 mM MgCl2, 10 μM GTP. A last step of gel filtration with Superdex 75 16/600 (GE Healthcare) was performed with a buffer containing 25 mM Hepes pH 7.5, 50 mM NaCl, 1mM MgCl2, 2 mM DTE, 10 μM GTP. 6x-His-KIF20A-529-665 was purified as in ref. [49]. GST-GFP was a kind gift of the recombinant antibody platform (Institut Curie).

Solid-phase assays were carried out as in ref. [34]. Briefly, 96-well plates (Nunc) were coated with 10 μg of either GST-GFP, GST-Myosin II-1148-1652, GST-RAB6A:GTP, and 0–40 μg recombinant 6xHis-RAB6:GTP or 6xHis-KIF20A-529-665 proteins in a buffer containing 20 mM Hepes, pH 7.5, 100 mM NaCl, and 5 mM MgCl2. Protein interaction was revealed by anti-6 × His antibodies (1:1000), using 100 μl TMD (BD Biosciences) as a chromogenic substrate at room temperature for 10 min. The reactions were stopped by the addition of sulphuric acid. Absorbance was measured at 450 nm in a plate reader.

**Purification, crystallization, and structure determination of RAB6:KIF20A-603-665 complex**. The KIF20A fragment (603–665) was cloned in pGEXII plasmid, containing a N-terminal GST-tag followed by a rTEV protease cleavage site. RAB6A-Q72L fragment (8–195) was cloned in a modified pACYC plasmid with an introduced rTEV cleavage site after the N-terminal His-tag. The complex was produced by co-expression of GST-KIF20A-603–665 and His-RAB6A-Q72L-8-195 in *E. coli* BL21 (DE3). The complex was purified using Glutathione Sepharose™ 4B in a batch mode. The complex was eluted by rTEV cleavage and

subsequently purified by Ni-NTA. Extra RAB6A-Q72L-8-195 was added to the purified complex mix to ensure saturation of KIF20A-RBD with the binding partner.

The complex at concentration of 10 mg/ml was crystallized in 100 mM Hepes pH 7.4, 200 mM ammonium sulfate, 24% (w/v) PEG 8 K, 2% (v/v) isopropanol in hanging drop geometry at 17 °C. A 2Å resolution data set was collected at the European Synchrotron Radiation Facility (ESRF) synchrotron (beamline ID29). The structure was determined by molecular replacement method using Molrep[52] and RAB6 structure (PDB ID 2GIL[53]) as a search model. The structure was iteratively rebuilt using Coot[54] and refined using Phenix[55] (Supplementary Table 1). The atomic coordinates and structure factors of the structure of the RAB6:KIF20A:RBD complex has been deposited in the Protein Data Bank (www. pdb.org) with accession number PDB ID 5LEF.

**Transport assays.** For vesicular stomatitis virus G protein (VSV-G)-trafficking experiments, HeLa cells were transfected with a plasmid allowing the expression of the thermosensitive mutant GFP-VSV-G tsO45 and incubated for 14 h at 40 °C. Transport to the Golgi complex and to the plasma membrane was induced by shifting cells at 32 °C for indicated times. Arrival of VSV-G at the plasma membrane was monitored by using a mouse monoclonal VG antibody directed against the exoplasmic domain of the protein, in the absence of detergent. Surface VSV-G was then revealed with a Cy3-labeled anti-mouse antibody. Cell to cell variation of the GFP-VSV-G expression levels were taken into account by measuring the ratio between plasma membrane VSV-G (surface labeling described in the Transport assay section) to the total VSV-G (surface + intracellular) after Z-projection of each color channels.

**Statistical analysis.** All data were generated from cells pooled from at least three independent experiments represented as (n), unless mentioned, in corresponding legends. Statistical data were presented as means ± standard error of the mean. Statistical significance was determined by Student's t test for two or three sets of data using Excel, no sample was excluded. Cells were randomly selected. Only P value <0.05 was considered as statistically significant.

**Data availability.** The authors declare that all relevant data supporting the findings of this study are available within the paper (and its Supplementary information file). Any raw data can be obtained from the corresponding author (S.M.-L.) on reasonable request.

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

## Acknowledgements

We are grateful to Franck Perez and Ana-Maria Lennon-Duménil (Institut Curie) for generous gifts of reagents; Ana-Maria Lennon-Duménil, Yohanns Bellaïche, and Franck Perez for critical reading of the manuscript and insightful discussions; Carina Santos, Laura Picas and Jean-Baptiste Brault and the Molecular Mechanisms Transport lab for productive discussions. The authors greatly acknowledge the Nikon Imaging Center at Institut Curie-CNRS, the PICT-IBiSA, member of the France-BioImaging national research infrastructure. This work was supported by an ERC (European Research Council) advanced grant (project 339847 "MYODYN") and an ARC grant (SL220120605302). We acknowledge the recombinant protein and antibody platform of the Institut Curie (http://umr144.curie.fr/en/plateform/protein-and-antibody-laboratory-001279) for the production of human recombinant antibodies against Giantin, MHC, and tubulin. The Goud and Houdusse teams are member of Labex CelTisPhyBio (11-LBX-0038) and Idex Paris Sciences et Lettres (ANR-10-IDEX-0001-02 PSL). We thank the beamline scientists of ID29 (ESRF synchrotron) for excellent support during data collection. O.P. was supported by an ARC post-doctoral fellowship. A.H. was supported by grants from the CNRS, ANR-10-MotoRab, ARC PJA 20151203285, ANR15-CE13-0016-01, Ligue Nationale Contre le Cancer.

## Author contributions

S.M.L. carried out the experiments presented in Figs. 1, 2a,d,e, 3, 4d–g, 5, 6b, Supplementary Figs. 1–7 with the help of S.B.; H.B. the experiments in Fig. 4h and constructs in Supplementary Figs. 3, 4; A.D. the experiments in Figs. 2c, 6a; V.F. helped for microscopes development and image analysis; G.B. and O.P. determined the X-ray structure with the help of R.B.; O.P., G.B. and A.H. analysed the structure; C.B. and C.G. designed, synthesized and developed paprotrain and BKS0349; S.M.L., A.E., A.H., B.G., O.P. conceived the project; S.M.L., H.B., A.D., A.E., A.H., B.G., O.P. designed and interpreted the experiments; S.M.L., O.P., A.E., A.H., B.G. wrote the manuscript. S.M.L., H.B., O.P., S.B., A.D., V.F., C.G., C.B., A.H., A.E. and B.G. edited the manuscript. B.G and A.H. secured funding.

## Additional information

**Competing interests:** The authors declare no competing financial interests.

