## [Peer Review File · Nature Communications]

Reviewers' Comments:

Reviewer #1 (Remarks to the Author)

The Miserey-Lenkei et al. manuscript is interesting and yields significant new information on vesicle/tubule formation and the molecular information on the formation of a Rab6/myosin/Kif20A complex. In many ways, the molecular information on the Rab6/myosin/Kif20A interactions is stronger than the more cellular data about tubular extensions and vesicle formation.

Major Points

Some important questions about the cellular data should be raised. How much overexpressed is tagged Rab6 that is used as the marker for much of the Golgi dynamics? There is no explicit data on the point. Interactions may be stabilized through overexpression and the point deserves to be addressed in the text. Rab6A' is the tagged Rab. Does RabA show the same interactions? That might be inferred from the fact that the Rab siRNA used was directed at both Rab6A and Rab6A'. Kif20A has been proposed to traffic Golgi vesicles retrograde in the direction of the ER. What is the directionality of the vesicles? At the resolution of the images, it is impossible to infer that the 3 molecule complex actually define a "hot" spot. There appears to be localization elsewhere. What defines a hot spot remains an open question and that point should be explicit. The relationship of flat areas and extremities in a Golgi ribbon to underlying Golgi structure is a speculation.

From looking at the movies, keeping track of what is exactly a hot spot has its difficulty. There are many events happening in each frame. I would urge the Journal to post on its website a full raw data copy of the image set that becomes Movie Supplement 1. That is the only way that one has a sufficient quality image set to play carefully the images over and over again and hence to think through what is being shown.

Minor Point

The first paragraph of the Results section needs to be rewritten. The sentences that state various numbers are out of order. First is stated 1-2 vesicles exit the Golgi, then it is stated that gives 12 vesicles per minute and finally it is stated that there are 6 hot spots. The order should be something like: a) 1-2 vesicles, b) 6 hot spots, and c) a total of 12 vesicles per minute per the scored Golgi area.

Reviewer #2 (Remarks to the Author)

It has been previously shown by the authors that the interaction between rab6 and myosin is critical for the fission of rab6-positive transport carriers from the Golgi/TGN. It has also been previously shown that rab6 and myosin interact with kif20a. In this study the authors show that kif20a is involved in, or required for, the fission of rab6-positive transport carriers, and that kif20a, rab6 and myosin form a complex.

They also report the existence of preferential sites for the formation of rab6-positive carriers from the Golgi/TGN, which they refer to as fission hot spots.

My comments are as follows.

The conclusion the kif20a is involved in fission is most probably correct, but it is not rigorous, as the experiment on which it is based rely only on the use of a chemical inhibitor (very unlikely to be selective) and of few siRNAs, which are well know to give frequently off target effects.

Moreover, given that kif20a was already know to interact with rab6 and myosin, the level of novelty of the finding that kif20a plays a role in the rab6 and myosin-dependent fission of these

carriers is modest, and does not advance much our understanding of carrier fission without some further degree of mechanistic analysis. In this regard, there are several questions that might be raised, for example: in what way is the fission-inducing activity of kif20a related to the motor properties of this protein; is the kif20a, rab6 and myosin complex physically or functionally related to other fission promoting molecular complexes known to act in post Golgi traffic; where is kif20a located in the tubular carrier emerging from the TGN: fission can occur all along the length of these carrier and not only, in fact, not frequently, at the base of the carriers, where kif20a appears to be located, etc.. Addressing these and other questions, and in particular the problem of role of the motor activity of kif20a in fission (which should also be tested more rigorously) would help to reach the necessary degree of novelty and completeness. In general, without an analysis of the mechanistic role of kif20a in fission, the study remains sort of preliminary, and does not reach the degree of significance that in my view would be required for publication in a visible journal.

The observation of preferential sites for carrier formation—referred to by the authors as fission hot spots—is potentially important, but, again, it remains preliminary at this stage. Moreover it is not clear how the hot spots issue relate to kif20a-dependent fission. The hot spots are sites where carriers form, where the key role is certainly played by the budding mechanism, and not by fission. Since kif20a rab6 and myosin do not seem to be involved in budding, it is not clear why the hot spots issue should be central in a study about fission.

A general problem is that the language is often unclear, and the claims the authors make are not easy to understand. For example, the Discussion starts with the statement: ... the main finding of this study is the existence of preferential sites for the formation of rab6-positive post-Golgi secretory vesicles on Golgi/TGN membranes... In view of the above, this sounds confusing.

Altogether, this manuscript presents some interesting and potentially important observations, which, however, in my view deserve to be developed much further before they can perhaps be considered for publication in a visible journal such as NatComm.

Reviewer #3 (Remarks to the Author)

The paper is a novel and interesting study of Golgi vesicle formation. The work follows previous studies of the role of Rab6 and motor proteins, and uses diverse methodology to understand the molecular/cellular basis for vesicle budding from Golgi membranes. The techniques and data interpretation are generally sound, and the results are sufficiently novel to merit publication. However, several issues need to be addressed to clarify the significance of the findings. Comments are listed in no particular order, and include typos:

- please clarify calculation - Over 60 sec movies, about 1-2

RAB6-positive vesicles were observed exiting the Golgi complex, giving a total of 12.5 {plus minus} 1.3 vesicles per minute per "hot spot" (n=13 Golgi) - unclear about this calculation, i.e. 1 vesicle per 60 seconds would be 1 vesicle per minute, can this calculation be clarified? How many seconds (on average) for vesicle formation at a hotspot?

Fig 3B, I assume 'MW' is Molecular weight in kDa? Should include this label

Fig 4 D legend, typo in 2nd sentence, 'fragment or or ...'

in Fig 3C, this key data is interpreted as revealing less myosin at Golgi, but very difficult to evaluate as reviewer, and quantification of fluorescence reduction can be problematic... what about protein quantification from Golgi extracts?

- issue of recruitment of myosin to Golgi - in siRNA knockout of Rab6, still plenty of KIF20A at Golgi (Fig 4F). Why is this? How efficient is siRNA against Rab6 (i.e. level of protein expression)? Is myosin still found at Golgi in siRNA KO of Rab6? In order to understand the molecular basis for budding of Golgi vesicles, it is important to clarify the steps that enable localization of the molecules

- the network of interactions between Rab6, myosin tail, and kinesin have been characterized by immunoprecipitations and yeast two-hybrid, indicating a complex set of binary and possibly ternary interactions at Golgi membranes. However, the current data (as presented) are vague, and do not provide a clear and insightful mechanistic view of the vesicle budding process. One problem is that the experiments are qualitative, and no quantitative affinities are presented. It would be useful to know the relative affinities in order to understand what are the most important and high-affinity interactions, eg, what is the equilibrium binding affinity of KIF-RBD and Rab6? KIF and myosin? Do the two sites for myosin on KIF act independently, or synergistically?

- is the disulfide bridge in kinesin tail necessary for Rab6 binding? Can this be shown by mutagenesis and binding/affinity and/or functional studies? Or were the sulfhydryls oxidized during purification of the isolated fragment of kinesin in vitro? It would be highly unusual mechanism, as disulfide bonds are rare inside cells

- regarding the interpretation of the mutagenesis/binding experiments, it is also possible that Rab6 and myosin bind to KIF20 at the same site, but in different ways, such that the tryptophan mutations do not affect myosin binding (in the region 529-665 of KIF20)

- data in Fig 4G suggests Rab6 is required for myosin-kinesin complexes. Is there simultaneous binding of Rab6 to two effectors, or is there independent recruitment (and thus, close spatial proximity) enabling myosin-kinesin binding? Again, it would be useful to have quantitative measurements of affinity (as indicated above) to shed light on these issues.

- the citations as superscript numbers are confusing in places, eg. Discussion 'Myosin II (n= 5, 6), that of KIF20A (n= 5, Fig. 2) or RAB6 depletion (n=4, 6). Here, and in other places throughout manuscript, it might be better to use 'ref.#' following a number.

- it is somewhat counter-intuitive and confusing to readers that a key finding is an additional binding site for MyoII on KIF20 (thus 2 total) using yeast two hybrid. These direct binding assays are independent of Rab6. However, in a cellular context, Rab6 appears crucial for enabling myosin and kinesin interactions. Clarification of these issues and articulation of a 3D-model for vesicle budding involving Rab6 and the multi-domain myosin and kinesin proteins would greatly enhance the quality of the paper for a general audience.

RESPONSE TO REVIEWERS

We thank the reviewers for their constructive remarks and provide below a point-by-point response to their comments.

REVIEWER #1

" How much overexpressed is tagged Rab6 that is used as the marker for much of the Golgi dynamics? There is no explicit data on the point. Interactions may be stabilized through overexpression and the point deserves to be addressed in the text."

We quantified by western blotting the amount of overexpressed GFP-RAB6 and endogenous RAB6. We found that both present similar level of expression. This result is presented in a new version of Fig. S1 (Fig. S1A).

Of note, GFP-RAB6 expressing cell line that we used in this study is the same as the one used in our previous work on fission (Miserey-Lenkei et al., NCB, 2010).

" Rab6A' is the tagged Rab. Does RabA show the same interactions? That might be inferred from the fact that the Rab siRNA used was directed at both Rab6A and Rab6A'."

We previously showed that myosin II forms a complex with GTP forms of RAB6A and RAB6A', and that both RAB6A and RAB6A' contribute to the recruitment and/or maintenance of myosin II at the Golgi (Miserey-Lenkei et al, NCB 2010). This is why we used in this study siRNAs that target both isoforms.

" Kif20A has been proposed to traffic Golgi vesicles retrograde in the direction of the ER. What is the directionality of the vesicles? "

We indeed proposed that KIF20A was functionally involved in the Golgi to ER retrograde transport of RAB6-positive vesicles (Echard et al, Science 1998), thus toward plus ends of microtubules, but the exact step at which this kinesin was involved was not defined. Since then, it was found that about 80% of RAB6-positive vesicles contain secretory cargos such as VSV-G and NPY (neuropeptide Y), which move from the Golgi to the plasma membrane (Grigoriev et al., Dev Cell 2007), thus again toward plus ends of microtubules. These results have been confirmed in our laboratory using VSV-G and other cargos (Miserey-Lenkei et al, NCB 2010, Fig. 7, and unpublished results). The transport at long distances of RAB6-positive vesicles toward the MT plus ends are now known to be mediated by the kinesin KIF5B. Importantly, RAB6 and Myosin II control both the fission of RAB6/secretory vesicles and the fission of RAB6/retrograde vesicles (Miserey-Lenkei et al, NCB 2010). This explains why RAB6 is involved in the traffic both from the Golgi to ER and from the Golgi to the plasma membrane.

In the present manuscript, we reveal the function of KIF20A at the Golgi by implicating

this kinesin in the fission of RAB6-positive vesicles. This explains why KIF20A was functionally involved in the Golgi to ER retrograde traffic in the initial study (Echard et al, Science 1998).

" At the resolution of the images, it is impossible to infer that the 3 molecule complex actually define a "hot" spot. There appears to be localization elsewhere. What defines a hot spot remains an open question and that point should be explicit. The relationship of flat areas and extremities in a Golgi ribbon to underlying Golgi structure is a speculation."

KIF20A, Myosin II and RAB6 are indeed localized to all over Golgi/TGN membranes. We define a "hot spot" the place where the three proteins can be detected together. The experiment presented in Fig. 2E was done to confirm that these dots of triple co-localization correspond to fission "hot spots". This is stated in the text: "Following KIF20A inhibition with paprotrain, dots containing the three proteins were found at the base of membrane tubes, suggesting that these dots correspond to the site of fission (Fig. 2E)". Finally, the relationship of flat areas and extremities in a Golgi ribbon and the location of the fission hot spots is only an experimental observation. We agree with the reviewer that we referred to "flat" vs. "extremities" of the Golgi ribbon at the resolution of the optic resolution, not at the ultrastructural level. We now clarified the text by indicating that "At the optical microscopy resolution, the Golgi fission "hot spots" are seen at the extremities rather than at the flatter regions of the Golgi."

" From looking at the movies, keeping track of what is exactly a hot spot has its difficulty. There are many events happening in each frame. I would urge the Journal to post on its website a full raw data copy of the image set that becomes Movie Supplement 1. That is the only way that one has a sufficient quality image set to play carefully the images over and over again and hence to think through what is being shown."

To facilitate the observation and interpretation of the movie S1, we now show all corresponding images in a new version of Fig. S1 (Fig. S1B). In addition, we will provide to the journal the raw data.

Minor Point

"The first paragraph of the Results section needs to be rewritten. The sentences that state various numbers are out of order. First is stated 1-2 vesicles exit the Golgi, then it is stated that gives 12 vesicles per minute and finally it is stated that there are 6 hot spots. The order should be something like: a) 1-2 vesicles, b) 6 hot spots, and c) a total of 12 vesicles per minute per the scored Golgi area."

The first paragraph has been rewritten to clarify the actual measurements and calculations, as suggested.

"Over 60 sec movies, we observed the existence of 6.4 ± 0.4 fission "hot spots" per Golgi (n=13 Golgi) and measured a total of 12.5 ± 1.3 vesicles per minute exiting the Golgi complex at fission "hot spots" (n= 13 Golgi). This indicates that 1-2 RAB6-positive vesicles exit the Golgi complex at fission "hot spots" per minute."

REVIEWER #2:

"The conclusion the kif20a is involved in fission is most probably correct, but it is not rigorous, as the experiment on which it is based rely only on the use of a chemical inhibitor (very unlikely to be selective) and of few siRNAs, which are well know to give frequently off target effects. "

The defect of fission observed following KIF20A inhibition was investigated by two different approaches: 1/ the use of a KIF20A chemical inhibitor, paprotrain (Fig. 2A and 2B) and 2/ the use of a specific siRNA (Fig. S2B). The selectivity of paprotrain has been tested (Tcherniuk et al, Angew. Chem. Int. Ed, 2010): the ATPase activity of a panel of 12 kinesins is not inhibited by paprotrain. Importantly, paprotrain does not inhibit the two closest related KIF20A kinesins MKLP-1 and MPP1 (M-phase phosphoprotein 1). The specificity of siRNA sequence targeting KIF20A used in this study has been validated by us and others (Neef et al, JCB, 2003). Furthermore, both treatments result in the appearance of binucleated cells (Fig. S2A), a characteristic phenotype similar to that observed following KIF20A inhibition or depletion (Hill et al, EMBO J, 2000; Neef et al, JCB, 2003).

In the revised version of the manuscript, we performed additional experiments to validate the specificity of fission defects observed after KIF20A inhibition:

1/ The biotech "Biokinesis" with which we collaborate is developing derivatives of paprotrain to formulate new compounds presenting a higher affinity for KIF20A (around 25 times higher affinity). We have tested one of this compound named BKS0349 (Fig. S2B). Similar experiments than those performed with paprotrain show that the treatment of cells with BKS0349 induce like paprotrain, the formation of long membrane tubules connected to the Golgi complex and a strong reduction in the number of RAB6-positive vesicles in the cytoplasm (Fig. S2B). These results further validate the involvement of KIF20A in the process.

2/ We now overexpressed in HeLa cells several truncated versions of KIF20A: GFP-KIF20A-25-665, GFP-KIF20A-529-665, mCherry-KIF20A-796-887, GFP-KIF20A-529-887 (Fig. S3A). These constructs are recruited to the Golgi complex (Fig. S3B) and behave as dominant negative mutants since there overexpression leads to a decrease in endogenous KIF20-Golgi associated staining and to the appearance of bi-nucleated cells (quantifications are given in the legend of Fig. S3 and in Fig. S3C). These truncated versions behave as dominant negative mutants likely because they all contain the coiled-coil domain involved in the homodimerization of KIF20A and hence could inhibit the interaction of KIF20A with specific partners.

In the revised version of the manuscript, we then tested the effect on fission of these dominant negative mutants. mCherry-RAB6 expressing HeLa cells were co-transfected with GFP-KIF20A-25-665, GFP-KIF20A-529-665, GFP-KIF20A-529-887 or GFP-RAB6 expressing HeLa cells were co-transfected with mCherry-KIF20A-796-887 and imaged by spinning-disk confocal time-lapse videomicroscopy (Fig. S3D). Expression of all constructs leads to fission defects: the appearance of tubes connected to the Golgi complex unable to fission and a decrease in the total number of vesicles (Fig. S3D-E).

In conclusion, several and independent approaches indicate that KIF20A is involved in the fission of RAB6-positive vesicles from the Golgi apparatus.

" Moreover, given that kif20a was already know to interact with rab6 and myosin, the level of novelty of the finding that kif20a plays a role in the rab6 and myosin-dependent fission of these carriers is modest, and does not advance much our understanding of carrier fission without some further degree of mechanistic analysis. In this regard, there are several questions that might be raised, for example:

in what way is the fission-inducing activity of kif20a related to the motor properties of this protein; "

We could not genetically address whether the motor activity of KIF20A is involved in the fission process. Indeed KIF20A overexpression bundles microtubules and affects Golgi organization (Echard et al., Science, 1998; S. Miserey-Lenkei, unpublished observations). We thus could not introduce a point mutation that would mimic a rigor mutant and check the effect on fission of this mutant.

However, paprotrain is a well documented inhibitor of the basal and microtubule-stimulated ATPase activity of KIF20A (Tcherniuk et al., Angew. Chem. Int. Ed, 2010). Paprotrain is an ATP uncompetitive inhibitor with a calculated K_i at the micromolar range. Mechanistically, paprotrain does not inhibit the binding of KIF20A to microtubules (Tcherniuk et al., Angew. Chem. Int. Ed, 2010). Based on these results we conclude that the motor property of KIF20A is involved in the fission process.

" is the kif20a, rab6 and myosin complex physically or functionally related to other fission promoting molecular complexes know to act in post Golgi traffic; "

This is an interesting question that is going beyond the scope of this study but should be addressed in future experiments. As said above, the majority of RAB6-positive vesicles contain the secretory marker VSV-G that was used in the studies that identified other fission promoting molecular complexes such as CtBP1-S/BARS, members of the 14-3-3 family, Phosphoinositide kinase PI4KIII β and lysophosphatidic acid acyltransferase δ (Bonazzi et al, NCB 2005; Valente et al, NCB 2012; Pagliuso et al; Nat Comm, 2016). It is thus quite possible that KIF20A/RAB6/Myosin II complexes are physically or functionally related to these complexes. We added a sentence in the discussion to highlight this point.

"where is kif20a located in the tubular carrier emerging from the TGN: fission can occur all along the length of these carrier and not only, in fact, not frequently, at the base of the carriers, where kif20a appears to be located, "

After paprotrain treatment (Fig. 2E), KIF20A/RAB6/Myosin II are found co-localized at the base of the tube. However, we could also observe KIF20A-labelling associated along a GFP-RAB6-positive tube emanating from the Golgi (see figure below). These profiles are quite rare, this is the reason why we did not show them in the submitted version of the paper. It is possible that KIF20A is present along the length of all carriers but below

detection levels in many cases.

"The observation of preferential sites for carrier formation– referred to by the authors as fission hot spots - is potentially important, but, again, it remains preliminary at this stage. Moreover it is not clear how the hot spots issue relate to kif20a-dependent fission. "

Golgi fission hot spots are defined as specific areas where RAB6-positive vesicles exit the Golgi complex. A good evidence that preferential sites for carrier formation correspond to fission "hot spots" is presented in Fig. 1B (Blebbistatin wash-out). To relate KIF20A-dependent fission to Golgi fission hot spots, we tried to perform similar experiments after paprotrain wash-out. Unfortunately trafficking did not resume after incubation of cells with this drug followed by wash-out.

At Golgi fission hot spots, the 3 identified players, RAB6, KIF20A and Myosin II, are found co-localized. Following KIF20A inhibition with paprotrain, spots of co-localization between the three proteins are found at the base of membrane tubes, clearly suggesting that these dots correspond to the site of fission (Fig. 2E).

These Golgi fission hot spots are localized at the base of growing microtubules from Golgi membranes as indicated by the co-localization between KIF20A and growing microtubules (Fig. 6A) as well as the triple co-localization between RAB6, Myosin II and GCC185 on Golgi membranes (Fig. 6B).

"The hot spots are sites where carriers form, where the key role is certainly played by the budding mechanism, and not by fission. Since kif20a rab6 and myosin do not seem to be involved in budding, it is not clear why the hot spots issue should be central in a study about fission."

The fission process that we observe in our movies is very rapid, around 1 sec (see also 1st answer to Reviewer #3). This means that the budding and fission processes are in fact intimately linked.

We agree with the reviewer that it is likely that RAB6/KIF20A/Myosin II are not involved in the budding process. The budding process could be driven by Myosin 1B (Almeida, NCB, 2011). The interplay between Myosin 1B and RAB6/KIF20A/Myosin II is an interesting question that is currently addressed in our laboratory.

A general problem is that the language is often unclear, and the claims the authors make are not easy to understand. For example, the Discussion starts with the statement: ... the

main finding of this study is the existence of preferential sites for the formation of rab6-positive post-Golgi secretory vesicles on Golgi/TGN membranes... In view of the above, this sounds confusing.

We apologize for the inconvenience. The text has been corrected to take into account this comment.

REVIEWER #3

“- please clarify calculation - Over 60 sec movies, about 1-2 RAB6-positive vesicles were observed exiting the Golgi complex, giving a total of 12.5 {plus minus} 1.3 vesicles per minute per "hot spot" (n=13 Golgi) - unclear about this calculation, i.e. 1 vesicle per 60 seconds would be 1 vesicle per minute, can this calculation be clarified? How many seconds (on average) for vesicle formation at a hotspot? “

The first paragraph has been rewritten to clarify the actual measurements and calculations, as suggested.

“Over 60 sec movies, we observed the existence of 6.4 ± 0.4 fission “hot spots” per Golgi (n=13 Golgi) and measured a total of 12.5 ± 1.3 vesicles per minute exiting the Golgi complex at fission “hot spots” (n= 13 Golgi). This indicates that 1-2 RAB6-positive vesicles exit the Golgi complex at fission “hot spots” per minute.”

Concerning the time needed for vesicle formation, as shown in Fig. 1A (example in region#1, between time 3-4, example in region#5, between time 4-5), this event takes usually 1 sec. This is now stated in the text: " The formation of a vesicle usually takes 1 sec (Fig. 1A; example in region#1, between time 3-4; example in region#5, between time 4-5). "

“Fig 3B, I assume 'MW' is Molecular weight in kDa? Should include this label”

This label has been included in the Figure legend.

“Fig 4 D legend, typo in 2nd sentence, 'fragment or or ...'”

The typo has been corrected.

“in Fig 3C, this key data is interpreted as revealing less myosin at Golgi, but very difficult to evaluate as reviewer, and quantification of fluorescence reduction can be problematic... what about protein quantification from Golgi extracts?”

Quantifications in the previous manuscript were carried out in HeLa cells in which myosin II staining at Golgi is not optimal. We repeated these experiments using rat cells and the mAD7 anti-myosin II antibody (Musch, JCB, 1997; Ikonen, JCS, 1997) that was used to quantify the amount of myosin II at the Golgi after RAB6 depletion (Miserey-Lenkei et al, NCB, 2010, Fig. 2). KIF20A depletion reduced by 35% the amount of Myosin II associated with the Golgi complex. These results are presented in a new Fig. 3C.

We agree that additional quantifications on Golgi extracts would be of interest. However, Golgi purification from cultured cells requires a large number of cells. These experiments are practically impossible when cells need to be treated with siRNAs.

“- issue of recruitment of myosin to Golgi - in siRNA knockout of Rab6, still plenty of KIF20A at Golgi (Fig 4F). Why is this? “

In the text, we state that RAB6 contributes to the recruitment of KIF20A on Golgi membranes. Other mechanisms may be implicated in KIF20A recruitment at the Golgi.

Alternatively, the pool of KIF20A present at the Golgi could be very stable. This point is now stated in the results section: "Since the recruitment of KIF20A on the Golgi only partially depends on RAB6, it is likely that additional mechanisms are implicated or alternatively that the Golgi-associated pool of KIF20A is very stable."

"How efficient is siRNA against Rab6 (i.e. level of protein expression)?"

RAB6 siRNA is very efficient as a nearly total depletion after 72h transfection with siRNA. The western-blotting showing the efficiency of RAB6 depletion is presented in a new version of Fig. S7A.

"Is myosin still found at Golgi in siRNA KO of Rab6?"

We previously reported that RAB6 depletion using siRNA (Miserey-Lenkei et al, NCB 2010; Fig. 2E) or MEFs from *RAB6* k/o mice treated with tamoxifen (Bardin et al, Biol Cell 2015; Fig. 6B) reduced by 50% the amount of myosin II associated with the Golgi complex.

"In order to understand the molecular basis for budding of Golgi vesicles, it is important to clarify the steps that enable localization of the molecules"

We have now presented a working model in a new version of Fig. 6 (Fig. 6C).

"- the network of interactions between Rab6, myosin tail, and kinesin have been characterized by immunoprecipitations and yeast two-hybrid, indicating a complex set of binary and possibly ternary interactions at Golgi membranes. However, the current data (as presented) are vague, and do not provide a clear and insightful mechanistic view of the vesicle budding process. One problem is that the experiments are qualitative, and no quantitative affinities are presented. It would be useful to know the relative affinities in order to understand what are the most important and high-affinity interactions, eg, what is the equilibrium binding affinity of KIF-RBD and Rab6? KIF and myosin? Do the two sites for myosin on KIF act independently, or synergistically?"

We agree with the reviewer and we have performed quantitative experiments. The results are presented in the new version of Fig. 4H.

We purified the GST-MyosinII-1148-1652 fragment, His-KIF20A-RBD-529-665, GST-RAB6 and His-RAB6. Using a previously described solid phase assay (Dambournet et al, NCB 2011; Fig. 1) we measured the relative binding affinities of RAB6-Myosin II, RAB6-KIF20A, and KIF20A-Myosin II interactions. We measured that RAB6 and Myosin II-1148-1652 bind to KIF20A-RBD-529-665 with similar affinities in the micromolar range. On the contrary, RAB6 binds to Myosin II-1148-1652 with around 5-times lower affinity. We thus conclude that RAB6 and Myosin II interact to KIF20A with the highest affinities suggesting that KIF20A is central for the interaction between RAB6 and Myosin II.

These results have been described in the text and in new Figure 4H:

" We next measured the relative binding affinities of RAB6 for Myosin II, RAB6 for KIF20A, and KIF20A for Myosin II using a previously described solid phase assay

(Dambournet, NCB, 2011). These experiments revealed that RAB6 and Myosin II-1148-1652 bind to KIF20A-529-665 with similar affinities in the micromolar range (Fig. 4H). On the other hand, RAB6 binds to Myosin II-1148-1652 with around 5-times lower affinity (Fig. 4H). The fact that RAB6 and Myosin II display the highest affinities for KIF20A suggests that KIF20A is central for the interaction between RAB6 and Myosin II. A likely scenario is that RAB6 first recruits KIF20A, which then binds to Myosin II (see model on Fig. 6C and discussion)."

Despite trying different purification methods, we did not manage to purify the KIF20A-796-887 fragment, and could thus not biochemically answer whether the two sites for myosin and KIF act independently, or synergistically. One explanation is that this fragment has a high pI (around 11) making it very difficult to purify.

However, we could test the effect of the mCherry-KIF20A-796-887 fragment in the fission process in HeLa cells. The results are presented in a new Fig. S3. mCherry-KIF20A-796-887 is recruited to the Golgi complex (Fig S3B) and behaves as a dominant negative mutant since its overexpression leads to a decrease in endogenous KIF20-Golgi associated staining and to the appearance of bi-nucleated cells (quantifications are given in the legend of Fig. S3 and Fig. S3C). Overexpression of this fragment leads to fission defects (Fig. S3D-E). Using this set of experiments, we could partially answer the question of the synergistic or independency of the two binding sites present on KIF20A over Myosin II binding. The results are presented in new Fig. S3. Overexpression of the KIF20A-529-887 domain which contains the two KIF20A binding sites does not show increase fission defects as compared to KIF20A-529-665 or KIF20A-796-887 fragments alone. We concluded from these results that the two binding sites present on KIF20A likely act independently over Myosin II.

These results are stated in the text:

"However, these two-binding sites likely act independently since overexpression of the KIF20A-529-887 domain does not lead to higher fission defects than the overexpression of the KIF20A-529-665 or KIF20A-761-887 domains alone (Fig. S3)."

"- is the disulfide bridge in kinesin tail necessary for Rab6 binding? Can this be shown by mutagenesis and binding/affinity and/or functional studies? Or were the sulfhydryls oxidized during purification of the isolated fragment of kinesin in vitro? It would be a highly unusual mechanism, as disulfide bonds are rare inside cells "

All residues that participate in the coiled-coil interactions also concur in positioning the cysteines on the parallel coiled-coil to be next to one another and thus facilitate the formation of the disulfide bond. The coiled-coil in this kinesin goes beyond the 603-665 region that was crystallized. It is most likely that the mutation of this cysteine would not disrupt the formation of the parallel coiled-coil. Whether stabilizing the coiled-coil with the cysteine would have a functional effect is unlikely but would require further investigation.

"- regarding the interpretation of the mutagenesis/binding experiments, it is also possible that Rab6 and myosin bind to KIF20 at the same site, but in different ways, such that the tryptophan mutations do not affect myosin binding (in the region 529-665 of KIF20)"

Yes, this is likely. Note also that the KIF20 dimeric coiled-coil has two Rab6 binding sites, one them might be used for Myosin II binding rather than for Rab6 binding. This point is

now stated in the results section: "This suggests that RAB6 and Myosin II may bind to the KIF20A dimeric coiled-coil (603-665) in different ways. Since this site has two RAB6 binding sites, one of them might be used for Myosin II binding rather than for RAB6 binding and the tri-partite complexes between KIF20A, RAB6 and Myosin II can still be formed."

"- data in Fig 4G suggests Rab6 is required for myosin-kinesin complexes. Is there simultaneous binding of Rab6 to two effectors, or is there independent recruitment (and thus, close spatial proximity) enabling myosin-kinesin binding? Again, it would be useful to have quantitative measurements of affinity (as indicated above) to shed light on these issues."

We answered this point by measuring the relative binding affinities (see answer previously).

"- the citations as superscript numbers are confusing in places, eg. Discussion 'Myosin II (n= 5, 6), that of KIF20A (n= 5, Fig. 2) or RAB6 depletion (n=4, 6). Here, and in other places throughout manuscript, it might be better to use 'ref.#' following a number."

We have clarified this point in the text, as suggested.

"- it is somewhat counter-intuitive and confusing to readers that a key finding is an additional binding site for MyoII on KIF20 (thus 2 total) using yeast two hybrid. These direct binding assays are independent of Rab6. However, in a cellular context, Rab6 appears crucial for enabling myosin and kinesin interactions. Clarification of these issues and articulation of a 3D-model for vesicle budding involving Rab6 and the multi-domain myosin and kinesin proteins would greatly enhance the quality of the paper for a general audience."

We have drawn a schematic which is presented in a new version of Fig. 6 (Fig. 6C). A detailed explanation of how we envisioned the formation of the Golgi fission hot spots is described in the legend of Fig. 6C.

Reviewers' Comments:

Reviewer #1 (Remarks to the Author)

This is a greatly improved manuscript compared to the original submission. It now concentrates on what is new and makes that clear. This manuscript provides substantial evidence as to how a Rab6-dependent hotspot for Golgi vesicular trafficking can be created by the interactions of Rab6, Kif20A, myosin IIA and a microtubule as an anchoring structure.

In the opinion of this reviewer, that is a significant advance. This reviewer, for once, will not quibble over the details of the experiments and how they might be made better. I have often done that in the past. I think here that the data are substantial and sufficient. I will, however, comment that the authors could have done the copyediting of the manuscript better. I assume that this will be done in detail as part of the journal process. But I must comment that the use of the word "lounge" in Methods when describing "a Roper Scientific laser lounge" did bring forth a laugh. Please use the word "launch" in the future.

Reviewer #2 (Remarks to the Author)

The authors have addressed my main concerns only partially.

In my first review, the opinion I expressed was ... given that KIF20A was already known to interact with Rab6 and myosin, the level of novelty of the finding is modest... the study is too preliminary... the work does not advance much our understanding of carrier fission... unless some further degree of mechanistic analysis is provided. Moreover, I asked a few questions that would have to be addressed to enhance the level of the manuscript.

One important question was addressed properly. My concern here was that the involvement of KIF20A in fission is not proven rigorously. In the revised version the authors provide sufficient evidence that KIF20A is indeed required for fission.

Other issues remain unclear, however.

One question was in what way the fission-inducing activity of KIF20A is related to the motor properties of this protein...

Here, the authors have done nothing to modify their previous version. They state they could not genetically address this point because KIF20A overexpression bundles microtubules. However, techniques of gene editing that do not require overexpression are now easily available and could have resolved this issue.

The authors also state that paprotrain is a well documented inhibitor of the basal and microtubule stimulated ATPase activity of KIF20A; and infer that, since paprotrain inhibits fission, the ATPase activity of KIF20A is necessary for fission. This is likely to be true, however, off-target effects cannot be excluded, so the conclusion of the authors is not rigorous. This is a pity, considering that the mechanism of action of KIF20A is a central problem here, and could have been tested. The proposal that a kinesin causes fission by acting as a motor is striking. One would like to see more solid evidence and more characterization and mechanistic analysis.

Another question was where KIF20A localizes in the tubular carrier emerging from the TGN. The literature indicates that fission can occur all along the length of these carriers and not only (in fact, not frequently) at the base of the carriers, where KIF20A can be visualized. If KIF20A acts in fission directly, it should be observed at the fission sites... Again, the authors do not add anything to their previous version. Where do the fission events occur in their hands? Is KIF20A present at

the fission sites? No information is provided.

A third point was that the observation of preferential sites for carrier formation (hot spots) might have nothing to do with fission. Repeated carrier formation most likely depends on properties of the carrier budding mechanism, rather than on the mechanism of fission. Yet the authors discuss the two phenomena as if they were intimately connected or even the same phenomenon. In my view, they are mixing two issues that are different and should be analyzed separately.

In conclusion this manuscript is stimulating but remains too preliminary and superficial to be recommended for publication. I would be willing to review it a third time if the author make an effort to improve it substantially along the lines indicate above.

Reviewer #3 (Remarks to the Author)

The paper submitted by Miserey-Lenkei et al reveals a role for myosin and kinesin interactions in the fission of Rab6-positive vesicles. It is significantly improved from the initial submission, with experiments that provide more insight into the molecular interactions in Golgi membranes.

In response too my suggestions, the following revisions have been performed:

- I appreciate Fig 4H, relating the binding affinities of the various proteins (myosin, kinesin and Rab6), it is a useful reference when trying to understand the mechanism underlying vesicle fission and sorting. KIF20A is clearly the key for the interactions between Rab6 and myosin.
- MyoII depletion in Rab6 siRNA knockouts is better quantified (Fig 3C)
- additional western blotting to show level of Rab6 depletion (Fig S7A)
- over expression of Kif20 binding sites for myosin, to show whether there may be synergistic or independent mechanisms in play (using a fission assay, Fig S3)

Following these revisions, the manuscript provides significant novel insight into a complex set of interactions in Golgi membranes. Therefore, its publication would be useful to a wide audience in cell and structural biology.

There are only a few minor corrections noted here:

- Fig 6C - panel on the left is too small, difficult to see the dotted window in the context of sub cellular compartments. Rather than expanding it, I would recommend removing the smaller figure, and simply labeling the TGN in the main figure, with possibly arrows/labels showing the relative positions of medial Golgi and PM
- in the pdf that I have, the crystallography table (S1) is truncated on the right side

RESPONSE TO REVIEWERS

REVIEWER #1

"This is a greatly improved manuscript compared to the original submission. It now concentrates on what is new and makes that clear. This manuscript provides substantial evidence as to how a Rab6-dependent hotspot for Golgi vesicular trafficking can be created by the interactions of Rab6, Kif20A, myosin IIA and a microtubule as an anchoring structure.

In the opinion of this reviewer, that is a significant advance. This reviewer, for once, will not quibble over the details of the experiments and how they might be made better. I have often done that in the past. I think here that the data are substantial and sufficient. I will, however, comment that the authors could have done the copyediting of the manuscript better. I assume that this will be done in detail as part of the journal process. But I must comment that the use of the word "lounge" in Methods when describing "a Roper Scientific laser lounge" did bring forth a laugh. Please use the word "launch" in the future."

We deeply thank reviewer #1 for her/his positive evaluation of our work.

As requested we replaced "lounge" by "launch" in the Methods section. We do apologize for this mistake.

REVIEWER #2

We thank Reviewer#2 for her/his constructive remarks and provide below a point-by-point response to her/his three main comments.

"The authors have addressed my main concerns only partially.

In my first review, the opinion I expressed was ... given that KIF20A was already know to interact with rab6 and myosin, the level of novelty of the finding is modest... the study is too preliminary... the work does not advance much our understanding of carrier fission... unless some further degree of mechanistic analysis is provided. Moreover, I asked a few questions that would have to be addressed to enhance the level of the manuscript.

One important question was addressed properly. My concern here was that the involvement of KIF20A in fission is not proven rigorously. In the revised version the authors provide sufficient evidence that KIF20A is indeed required for fission.

Other issues remain unclear, however.

One question was in what way the fission-inducing activity of KIF20A is related to the motor properties of this protein...

Here, the authors have done nothing to modify their previous version. They state they could not genetically address this point because KIF20A overexpression bundles microtubules. However, techniques of gene editing that do not require overexpression are now easily available and could have resolved this issue.

The also state that paprotrain is a well documented inhibitor of the basal and microtubule stimulated ATPase activity of KIF20A; and infer that, since paprotrain inhibits fission, the ATPase activity of KIF20A is necessary for fission. This is likely to be true, however, off-

target effects cannot be excluded, so the conclusion of the authors is not rigorous. This is a pity, considering that the mechanism of action of KIF20A is a central problem here, and could have been tested. The proposal that a kinesin causes fission by acting as a motor is striking. One would like to see more solid evidence and more characterization and mechanistic analysis."

In order to answer to Reviewer#2's concerns, in the previous revised version of the manuscript, we used two different specific inhibitors of KIF20A known to inhibit the ATPase activity of KIF20A (Tcherniuk et al, Angew. Chem. Int. Ed, 2010). Cells treated with these two inhibitors showed fission defects. In addition, as shown in Fig. S3 of the previous revised version, overexpression of a truncated form of KIF20A lacking the motor domain (KIF20A-529-887) also led to fission defects. We considered that these two independent sets of results were sufficient to conclude that the motor activity of KIF20A promotes the fission process.

As requested by Reviewer#2, to provide further genetic evidence (not based on inhibitors), we carried out additional rescue experiments to directly test the role of the ATPase activity of KIF20A.

Based on published structural information on kinesin and myosin motors, establishing an absolute requirement for the conserved lysine of the P-loop in nucleotide binding and motor activity (Logan, J. Mol. Biol 1993; Li, J. Biol. Chem. 1998; Kull, J. Muscle Res. Cell. Motil.1998), we designed the K165A mutation in KIF20A. We also optimized the transfection protocols to allow a low expression level of GFP-KIF20A or GFP-KIF20A-K165A, thus avoiding the KIF20A-dependent bundling of microtubules. GFP-KIF20A and GFP-KIF20A-K165A were partially localized to the Golgi area (Fig. S4A) and expressed at similar levels (Fig. S4B). GFP-KIF20A or GFP-KIF20A-K165A were expressed in control or KIF20A-siRNA treated cells and we measured the number of tubes and vesicles in these different conditions. As shown in Fig. S4C, GFP-KIF20A WT rescued the fission defects associated with KIF20A depletion. In contrast, GFP-KIF20A-K165A was unable to rescue these defects. Interestingly, GFP-KIF20A-K165A even had a dominant negative effect on fission when expressed in cells that have not been depleted in KIF20A. Altogether, these additional results demonstrate that the ATPase activity of KIF20A is required in the fission process.

These results have been incorporated in the results section (page 6-7) and in a new Fig. S4.

In the discussion section, we propose that the motor activity of KIF20A could be important to generate the tension necessary for vesicles fission and/or for the local organization of microtubules favouring the right positioning of the fission machinery.

« Another question was where KIF20A localizes in the tubular carrier emerging from the TGN. The literature indicates that fission can occur all along the length of these carriers and not only (in fact, not frequently) at the base of the carriers, where KIF20A can be visualized. If KIF20A acts in fission directly, it should be observed at the fission sites... Again, the authors do not add anything to their previous version. Where do the fission events occur in their hands? Is KIF20A present at the fission sites? No information is provided. »

In the previous version of this paper, we have shown that KIF20A is localized at the base of the emerging tubes obtained following fission inhibition (Fig. 2E). We also showed in the answer to Reviewer#2 images of cells where we could observe KIF20A along the tubes.

In order to answer to the Reviewer#2's question more precisely, we carried out additional experiments. Monitoring fission events by two colours live cell imaging is very challenging. We thus used another strategy. To demonstrate the involvement of Myosin II in the fission process, we have previously shown that after wash-out of the Myosin II inhibitor blebbistatin, bulges of membrane appeared along the unfissioned tubules. These bulges accumulate Myosin II and, after a few seconds, correspond to the site where fission occurs (Miserey-Lenkei et al., NCB 2010).

We carried out the same kind of blebbistatin wash-out experiments and looked whether KIF20A localized at the membranes bulges. As shown in new Fig. S5E, we observed an accumulation of KIF20A at membrane bulges enriched in GFP-RAB6. These results demonstrate that, when Myosin II is reactivated and fission occurs, KIF20A is present at the fission sites.

"A third point was that the observation of preferential sites for carrier formation (hot spots) might have nothing to do with fission. Repeated carrier formation most likely depends on properties of the carrier budding mechanism, rather than on the mechanism of fission. Yet the authors discuss the two phenomena as if they were intimately connected or even the same phenomenon. In my view, they are mixing two issues that are different and should be analyzed separately."

We do agree that budding and fission of RAB6-positive vesicles could be distinct mechanisms, in particular because we and others have previously shown that RAB6 depletion leads to an accumulation of budding profiles on Golgi membranes (Bardin et al., Biol. Cell 2015; Storrie et al., Traffic 2012). To clarify this point, we have added a few sentences in the discussion section (page 12).

"In conclusion this manuscript is stimulating but remains too preliminary and superficial to be recommended for publication. I would be willing to review it a third time if the author make an effort to improve it substantially along the lines indicate above."

We hope that the reviewer will recognize our efforts to improve our manuscript and will now recommend it for publication.

REVIEWER #3

"The paper submitted by Miserey-Lenkei et al reveals a role for myosin and kinesin interactions in the fission of Rab6-positive vesicles. It is significantly improved from the initial submission, with experiments that provide more insight into the molecular interactions in Golgi membranes.

In response too my suggestions, the following revisions have been performed:

- I appreciate Fig 4H, relating the binding affinities of the various proteins (myosin, kinesin and Rab6), it is a useful reference when trying to understand the mechanism

underlying vesicle fission and sorting. KIF20A is clearly the key for the interactions between Rab6 and myosin.

- MyoII depletion in Rab6 siRNA knockouts is better quantified (Fig 3C)*
- additional western blotting to show level of Rab6 depletion (Fig S7A)*
- over expression of Kif20 binding sites for myosin, to show whether there may be synergistic or independent mechanisms in play (using a fission assay, Fig S3)*

Following these revisions, the manuscript provides significant novel insight into a complex set of interactions in Golgi membranes. Therefore, its publication would be useful to a wide audience in cell and structural biology.

There are only a few minor corrections noted here:

- Fig 6C - panel on the left is too small, difficult to see the dotted window in the context of sub cellular compartments. Rather than expanding it, I would recommend removing the smaller figure, and simply labeling the TGN in the main figure, with possibly arrows/labels showing the relative positions of medial Golgi and PM*
- in the pdf that I have, the crystallography table (S1) is truncated on the right side"*

We deeply thank reviewer #3 for her/his positive evaluation of our work.

As requested:

- In Fig. 6C, we deleted the left panel and labelled the TGN, PM, Trans- and Cis-Golgi in the main figure.*
- We changed the disposition of Table S1 to avoid its truncation on the right.*

Reviewers' Comments:

Reviewer #2:

Remarks to the Author:

The authors have addressed most of the concerns.

After two detailed reviews, it is time to decide whether the manuscript should be accepted or not.

I think I can recommend this manuscript for publication in Nature Communications